# Hypoexcitability precedes denervation in the large fast-contracting motor units in two unrelated mouse models of ALS

María de Lourdes Martínez-Silva[1], Rebecca D Imhoff-Manuel[1], Aarti Sharma[2], CJ Heckman[3,4,5], Neil A Shneider[2], Francesco Roselli[6], Daniel Zytnicki[1], Marin Manuel[1,3]*

[1]Centre de Neurophysique, Physiologie et Pathologie, CNRS, Université Paris Descartes, Paris, France; [2]Center for Motor Neuron Biology and Disease, Department of Neurology, Columbia University, New York, United States; [3]Department of Physiology, Northwestern University, Feinberg School of Medicine, Chicago, United States; [4]Department of Physical Medicine and Rehabilitation, Northwestern University, Feinberg School of Medicine, Chicago, United States; [5]Department of Physical Therapy and Human Movement Science, Northwestern University, Feinberg School of Medicine, Chicago, United States; [6]Department of Neurology, Ulm University, Ulm, Germany

**Abstract** Hyperexcitability has been suggested to contribute to motoneuron degeneration in amyotrophic lateral sclerosis (ALS). If this is so, and given that the physiological type of a motor unit determines the relative susceptibility of its motoneuron in ALS, then one would expect the most vulnerable motoneurons to display the strongest hyperexcitability prior to their degeneration, whereas the less vulnerable should display a moderate hyperexcitability, if any. We tested this hypothesis in vivo in two unrelated ALS mouse models by correlating the electrical properties of motoneurons with their physiological types, identified based on their motor unit contractile properties. We found that, far from being hyperexcitable, the most vulnerable motoneurons become unable to fire repetitively despite the fact that their neuromuscular junctions were still functional. Disease markers confirm that this loss of function is an early sign of degeneration. Our results indicate that intrinsic hyperexcitability is unlikely to be the cause of motoneuron degeneration.

DOI: https://doi.org/10.7554/eLife.30955.001

*For correspondence:
marin.manuel@neurobio.org

Competing interests: The authors declare that no competing interests exist.

## Introduction

Intrinsic hyperexcitability of motoneurons has been suggested to contribute to their degeneration in amyotrophic lateral sclerosis (ALS)(reviewed in *Van Den Bosch et al., 2006*). Hyperexcitability would increase the discharge probability and thereby the calcium inflow, a key step in the excitotoxic process. In fact, changes in excitability occur very early in mutant superoxide dismutase 1 (mSOD1) mice, a standard model of ALS. In mSOD1 embryos, spinal motoneurons are hyperexcitable (*Pieri et al., 2003*; *Kuo et al., 2005*; *Martin et al., 2013*). In neonates, motoneuron excitability was found to be increased (hyperexcitable) in some studies (*van Zundert et al., 2008*; *Elbasiouny et al., 2010*; *Leroy et al., 2014*), unaltered in others (*Pambo-Pambo et al., 2009*; *Quinlan et al., 2011*), and in *Bories et al. (2007)*, a decrease in input resistance was reported, suggesting that spinal motoneurons became hypoexcitable. At a later time point, in adults, before the onset of degeneration, we reported that a substantial fraction of motoneurons lose their ability to elicit a repetitive

**eLife digest** Amyotrophic lateral sclerosis (ALS), also known as Lou Gehrig's disease, is a fatal disorder of the nervous system. Early symptoms include muscle weakness, unsteadiness and slurred speech. These symptoms arise because the neurons that control muscles – the motoneurons – lose their ability to make the muscles contract. Eventually, the muscles become paralyzed, with more and more muscles affected over time. Most patients die within a few years of diagnosis when the disease destroys the muscles that control breathing.

Muscles are made up of muscle fibers. Each motoneuron controls a bundle of muscle fibers, and the motoneuron and its muscle fibers together make up a motor unit. A single muscle contains hundreds of motor units. These consist of several different types, which differ in how many muscle fibers they contain, how fast those muscle fibers can contract, and how fatigable the muscle fibers are. In ALS, motoneurons become detached from their muscle fibers, causing motor units to break down. But what triggers this process? One long-standing idea is that motoneurons begin to respond excessively to commands from the brain and spinal cord. In other words, they become hyperexcitable, which ultimately leads to their death.

But some more recent studies of ALS suggest the opposite, namely that motoneurons become less active, or hypoexcitable. To distinguish between these possibilities, Martinez-Silva et al. took advantage of the fact that different types of motor unit break down at different rates in ALS. Large motor units containing fast-contracting muscle fibers break down before smaller motor units. By measuring the activity of motor units in two mouse models of ALS, Martinez-Silva et al. showed that large motoneurons are hypoexcitable. In other words, the motoneurons that are most vulnerable to ALS respond too little to commands from the nervous system, rather than too much.

Studies of specific proteins inside the cells confirmed that hypoexcitable motoneurons are further along in the disease process than other motoneurons. Hypoexcitability is thus a key player in the ALS disease process. Developing drugs to target this hypoexcitability may be a promising strategy for the future of this condition.

DOI: https://doi.org/10.7554/eLife.30955.002

discharge in response to sustained inputs (*Delestrée et al., 2014*), a clear manifestation of hypoexcitability. The remaining motoneurons, which still produced a repetitive discharge, retained normal excitability (*Delestrée et al., 2014*). Studies in motoneurons derived from human iPSCs confirmed this tendency for excitability to shift from a hyperexcitable (*Wainger et al., 2014*) to a hypoexcitable state (*Sareen et al., 2014*; *Devlin et al., 2015*; *Naujock et al., 2016*) during the course of the disease. Altogether these observations challenge the role of intrinsic hyperexcitability in neurodegeneration.

A salient feature in ALS is that the differential vulnerability of motoneurons depends on the physiological type of their motor unit. The most vulnerable and the first to degenerate are the motoneurons of the largest motor units that innervate the fast-contracting and fatigable muscle fibers (FF type), followed by the motoneurons of intermediate motor unit size, which innervate the fast-contracting and fatigue-resistant muscle fibers (FR type). Finally, the motoneurons of the smallest motor units, that mainly innervate the slow contracting and fatigue-resistant fibers (S type) are the most resistant and the least affected at the end-stage of disease (*Pun et al., 2006*; *Hegedus et al., 2008*). If membrane hyperexcitability were to cause motoneuron degeneration, then the most vulnerable motoneurons should display the strongest hyperexcitability, whereas the less vulnerable ones should display a moderate hyperexcitability if any. In a recent study, we demonstrated that only a subgroup of motoneurons are hyperexcitable (rheobase is lowered) at P6–P10 in mSOD1 mice. However, using a combination of electrophysiological, morphological and molecular criteria, these motoneurons were identified as S-type, that is the most resistant in ALS (*Leroy et al., 2014*). In sharp contrast, the intrinsic excitability of the most vulnerable motoneurons is unchanged at this age, arguing against the idea that early hyperexcitability contributes to selective degeneration.

Nevertheless, degeneration of the most vulnerable motoneurons starts several weeks later (the FF motor units lose their neuromuscular junctions around 60 days post-natal in the SOD1G93A model; *Pun et al., 2006*). How excitable, then, are the vulnerable motoneurons in the days that

precede their degeneration? We have previously shown that some motoneurons are hypoexcitable at this time point, but we were unable to determine if those motoneurons were vulnerable or resistant (*Delestrée et al., 2014*). Here, we perform a critical experiment to answer this question using intracellular recordings of spinal motoneurons in non-paralyzed, deeply anesthetized adult mice to investigate the electrical properties of individual motoneurons as well as the compound muscle action potential and the contractile properties of the innervated muscle fibers (*Manuel and Heckman, 2011*). By this method, we were able (1) to determine that every motoneuron under investigation retains functional neuromuscular junctions, indicating that its degeneration has not yet begun, and (2) to identify the physiological type of its motor unit. Our experiments revealed that motoneurons that failed to discharge repetitively in response to sustained inputs are distributed predominantly among the FF and the largest FR motor units. Furthermore, we asked whether this property was specific to the SOD1$^{G93A}$ model, or may generally distinguish vulnerable motoneurons in ALS. To do this, we repeated our experiments in a mouse model of FUS-ALS (*Sharma et al., 2016*) and found again that large motor units tend to become hypoexcitable over disease progression.

Combining in vivo electrophysiology and in situ hybridization, we have investigated the expression pattern of C-type lectin chondrolectin (*Chodl*) in WT mice since it has been proposed as a marker for Fast-type motoneurons (*Enjin et al., 2010*) that disappears over the course of ALS progression (*Wootz et al., 2010*). In WT mice, we found that *Chodl* is expressed in the FF motoneurons, as well as in the largest FR motoneurons, but not in the small FRs, matching the hypoexcitability pattern in SOD1$^{G93A}$ mice. Altogether, our data indicate that FR motoneurons constitute a heterogeneous population that differentially lose their ability to fire repetitively over the course of disease progression, a property that may underlie their differential vulnerability in ALS. Moreover, the expression patterns of p-eIF2α and p62 aggregates suggest that the hypoexcitable motoneurons are more advanced in the disease progression than those that still fire repetitively.

We conclude that in ALS mice, both the FF and the largest FR motoneurons are not hyperexcitable prior to their degeneration, but on the contrary, lose the fundamental property of firing repetitively.

## Results

### Motor units classification based on their contractile properties

We identified the physiological type of motor units from the Triceps Surae (TS) of WT (106 motor units) and SOD1$^{G93A}$ mice (94 motor units). Recordings were performed on 46 to 60 days old animals (WT: $54 \pm 5$ days; N = 45; SOD1$^{G93A}$: $52 \pm 5$ days; N = 36). In each motor unit, we performed intracellular recordings at the motoneuron soma, while, at the same time, recording the force developed at the muscle tendon. Motor unit type-identification rested on the contractile properties of the motor units evaluated by applying short pulse stimulations to the motoneuron to elicit motor unit twitches (*Figure 1A$_1$, B$_1$ and C$_1$*), as well as a series of short pulse trains (8 spikes at 40 Hz, repeated every second for 3 min) to elicit a long series of unfused tetani (*Figure 1A$_2$, B$_2$ and C$_2$*). *Figure 1* illustrates three representative examples of the force profiles recorded, while *Figure 2* shows that, despite the fact that the contractile properties (namely twitch amplitude, twitch contraction time and fatigue index) vary widely across the motor unit population, they clearly display some inter-relationships in both WT and SOD1$^{G93A}$ animals. As illustrated in *Figure 1*, one of the largest WT motor units in our sample (twitch amplitude 12.7 mN; *Figure 1A*) had a fast contraction time (10 ms) whereas the slowest WT motor unit (contraction time 33 ms; *Figure 1C*) developed a weak force (0.9 mN). However, the WT motor unit illustrated in *Figure 1B* was equally fast as the motor unit illustrated in *Figure 1A* (contraction time 11 ms) but developed a much smaller twitch force (4.3 mN). Overall, the contractile properties were remarkably similar between WT and SOD1$^{G93A}$ animals. In both types of mice, there is a clear population of motor unit that contracts slowly and that develops a small amount of force (*Figure 2A$_1$ and B$_1$*). We chose a cutoff value of 20 ms to separate those 'slow' motor units from the 'fast' contracting ones. Indeed, all the motor units with a contraction time longer than 20 ms developed a weak twitch force (all but one under 1.0 mN: WT: $0.7 \pm 0.5$ mN; [0.2–1.8 mN]; N = 8; SOD1$^{G93A}$: $0.4 \pm 0.3$ mN; [0.1–1.1 mN]; N = 12; *Figure 2A$_1$*). In contrast, the fast motor units (contraction time shorter than 20 ms) displayed a much wider range of twitch

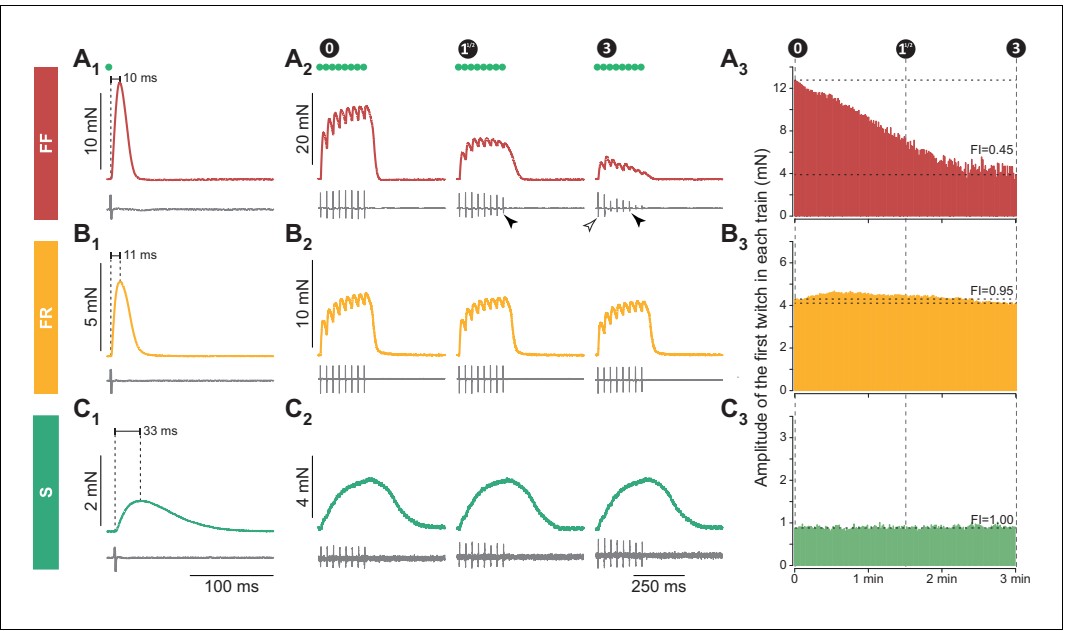

**Figure 1.** Examples of 3 motor units with different contractile properties. (**A**) FF motor unit. (**A₁**) Twitch (top trace), and motor unit action potential (MUAP, bottom trace) elicited by a spike generated in the cell body of the investigated motoneuron (green dot indicates the time the spike was generated). (**A₂**) Examples of unfused tetani recorded at the beginning (0 min) the middle ($1^{1/2}$ min) and the end (3 min) of the fatigue test. Note that, in this particular motor unit, the MUAP tended to decrease during the train (filled arrowhead), but the amplitude of the first MUAP stayed constant during the fatigue test (empty arrowhead). (**A₃**) Time course of the decline in amplitude of the first twitch in each train over the duration of the fatigue test. Horizontal dotted lines indicate the amplitude of the first twitch in the first train and the amplitude of the first twitch in the last train, used to calculate the Fatigue Index (FI; see Materials and methods). (**B**) FR unit, same arrangement as in A. (**C**) S motor unit, same arrangement as in A. Traces in $A_1$, $B_1$ and $C_1$ are averages of 5–10 sweeps.

DOI: https://doi.org/10.7554/eLife.30955.003

force (WT: 3.8 ± 5.4 mN; [0.1–29.8 mN]; N = 98; SOD1$^{G93A}$: 5.9 ± 10.3 mN; [0.1–53.1 mN]; N = 80; *Figure 2A₁*).

In addition, we measured the resistance to fatigue on a subset of 90/106 motor units in WT mice and 43/94 in SOD1$^{G93A}$ mice (*Figure 1*). In our sample, the fatigue indexes (measured using the first twitch of the trains; see Materials and methods) ranged from 1.00 (not fatigable at all), to 0.10 (fully fatigable) (0.90 ± 0.21; N = 90 in WT and 0.90 ± 0.23; [0.08–1.00]; N = 43 in SOD1$^{G93A}$ mice). As shown in *Figure 2A₂₋₃ and B₂₋₃*, the fatigue index correlated with the twitch contraction time and the twitch amplitude in both WT and SOD1$^{G93A}$ mice. Motor units with a contraction time longer than 20 ms were all fatigue resistant (fatigue index between 0.86 and 1.00 in both WT and SOD1$^{G93A}$ mice, *Figure 2A₂ and B₂*; see also the slow-contracting motor unit in *Figure 1C*). The motor units with a fast contraction time (<20 ms) spanned the whole range of fatigue indexes (between 0.10 and 1.0, *Figure 2A₂ and B₂*). However, all of the fatigable motor units (fatigue index below 0.5) developed a large initial amount of twitch force (e.g. 12.7 mN for the motor unit in *Figure 1A*). In sharp contrast, the vast majority of fast-contracting motor units that displayed a fatigue index above 0.5 developed less force (e.g. 4.3 mN for the motor unit in *Figure 1B*). Remarkably, with very few exceptions, we were able to distinguish fast contracting, fatigue resistant (FR) from fast-contracting, fatigable (FF) units a cut-off value of 8 mN (*Figure 2A₃ and B₃*). Taken together, we conclude that a motor unit can be classified solely on the features of the twitch to classify the motor units: S-type if contraction time ≥20 ms; FR if contraction time <20 ms and twitch amplitude <8 mN; FF if contraction time <20 ms and twitch force ≥8 mN). In WT mice, we classified 13 motor units as FF (12%), 85 as FR (80%) and eight as S (8%). In SOD1$^{G93A}$ mice, we classified 15 units as FF (16%), 65 as FR (71%) and 12 as S (13%). The proportions of motor units of each type found in WT vs. SOD1$^{G93A}$ mice were not statistically different ($\chi^2$ test p=0.28), and are consistent

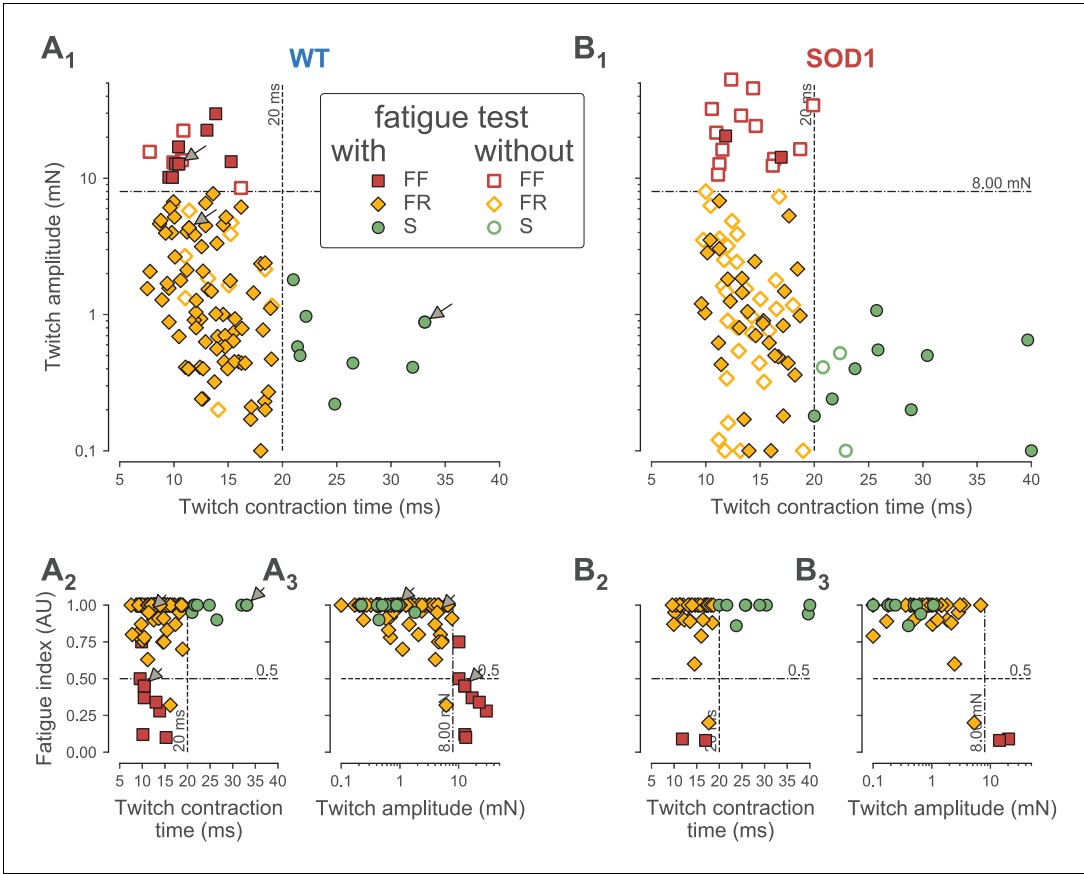

**Figure 2.** Classification of motor units. (A) Contractile properties of WT motor units. (A₁) Distribution of the twitch amplitude (logarithmic scale) vs. twitch contraction time. The motor units indicated by an arrow correspond to the three motor units of *Figure 1*. The vertical dashed line at 20 ms represents the limit between the fast and slow-contracting motor units. The horizontal dash-dotted line at 8 mN represents the limit between FR and FF motor units. The filled markers correspond to the motor units in which the fatigue index was measured, while the empty markers correspond to the motor units for which the fatigability was not measured. (A₂) Distribution of the Fatigue Index vs. the twitch contraction time. The dash-dotted line at 0.5 represents the limit between fatigue-resistant and fatigable motor units. (A₃) Distribution of the Fatigue Index vs. twitch amplitude (logarithmic scale). (B) Contractile properties of SOD1^G93A motor units. Same organization as in A.

DOI: https://doi.org/10.7554/eLife.30955.004

with the proportions of muscle fibers found in this muscle (*Bloemberg and Quadrilatero, 2012*). *Table 1* shows the contractile properties of the different types of motor units in WT and SOD1^G93A mice. We observed that FF motor units display a slightly longer contraction time in mutant mice. Since these units develop a similar amount of force as in control mice (*Table 1*), this might reflect a desynchronization of their muscle fibers that precedes denervation proper.

## Loss of function in the motoneurons innervating the largest motor units

Excitability of type-identified motoneurons was tested in response to a slow triangular ramp of current in 80 SOD1^G93A motoneurons and in 63 WT motoneurons. As we previously described (*Delestrée et al., 2014*), a substantial fraction of motoneurons was not able to discharge in response to the ramp, despite the fact that they were still able to fire at the onset of a square pulse (*Figure 3A*). In contrast, the other motoneurons discharged repetitively during the ramp (*Figure 3B*). This lack of repetitive firing is a manifestation of hypoexcitability of these motoneurons. Notably, this loss of function occurs before the denervation of the neuromuscular junctions as every cell in this study was able to activate its muscle fibers (as shown by the recording of a motor unit action potential at the surface of the muscle) and to produce a twitch in response to a spike elicited

**Table 1.** Contractile properties of the different types of motor units in WT and SOD1$^{G93A}$ mice.

For each property, we report the mean ± SD as well as the range (in brackets), and the sample size for each genotype, and for each physiological type (FF, FR and S). In addition, we report the values in the large motor units (FF and large FR with a twitch force larger than 1.3 mN), and the small motor units (small FR with a twitch force smaller than 1.3 mN and S). The column diff shows the result of a Mann-Whitney test between WT and SOD1$^{G93A}$ mice. The test could only be performed if both samples contained at least five motoneurons.

| Property | Motor unit type | WT | SOD1$^{G93A}$ | Diff. |
|---|---|---|---|---|
| Twitch amplitude | Large units | 6.4 ± 6.1 mN; [1.3–29.8 mN]; N = 54 | 10.7 ± 12.5 mN; [1.4–53.1 mN]; N = 42 | NS |
| | Small units | 0.6 ± 0.4 mN; [0.1–1.8 mN]; N = 52 | 0.6 ± 0.4 mN; [0.1–1.3 mN]; N = 50 | NS |
| | FF | 15.5 ± 6.1 mN; [8.5–29.8 mN]; N = 13 | 23.8 ± 12.8 mN; [10.6–53.1 mN]; N = 15 | NS |
| | FR | 2.0 ± 2.0 mN; [0.1–7.7 mN]; N = 85 | 1.8 ± 1.8 mN; [0.1–8.0 mN]; N = 65 | NS |
| | S | 0.7 ± 0.5 mN; [0.2–1.8 mN]; N = 8 | 0.4 ± 0.3 mN; [0.1–1.1 mN]; N = 12 | NS |
| Twitch contraction time | Large units | 12.1 ± 2.8 ms; [7.5–18.4 ms]; N = 54 | 13.2 ± 2.7 ms; [9.8–19.9 ms]; N = 42 | * (p=0.033) |
| | Small units | 16.2 ± 5.0 ms; [8.9–33.1 ms]; N = 52 | 17.4 ± 6.6 ms; [9.6–40.0 ms]; N = 50 | NS |
| | FF | 11.4 ± 2.4 ms; [7.8–16.2 ms]; N = 13 | 14.0 ± 3.0 ms; [10.6–19.9 ms]; N = 15 | ** (p=0.005) |
| | FR | 13.5 ± 3.0 ms; [7.5–19.0 ms]; N = 85 | 13.8 ± 2.6 ms; [9.6–19.0 ms]; N = 65 | NS |
| | S | 25.3 ± 4.8 ms; [21.0–33.1 ms]; N = 8 | 26.8 ± 6.8 ms; [20.0–40.0 ms]; N = 12 | NS |
| Fatigue Index | Large units | 0.8 ± 0.3; [0.1–1.0]; N = 40 | 0.7 ± 0.4; [0.1–1.0]; N = 13 | NS |
| | Small units | 1.0 ± 0.1; [0.7–1.0]; N = 50 | 1.0 ± 0.1; [0.8–1.0]; N = 30 | NS |
| | FF | 0.4 ± 0.2; [0.1–0.8]; N = 8 | 0.1 ± 0.0; [0.1–0.1]; N = 2 | - |
| | FR | 1.0 ± 0.1; [0.3–1.0]; N = 74 | 0.9 ± 0.2; [0.2–1.0]; N = 32 | NS |
| | S | 1.0 ± 0.0; [0.9–1.0]; N = 8 | 1.0 ± 0.0; [0.9–1.0]; N = 9 | NS |

DOI: https://doi.org/10.7554/eLife.30955.006

by a short pulse of current (*Figure 3A$_2$*). The loss of function was significantly more frequent in SOD1$^{G93A}$ motoneurons than in WT motoneurons (21% vs. 6%, exact Fisher test p=0.01, *Figure 3C*). Moreover, non-firing motoneurons were not distributed homogeneously in all physiological types of motor units, but were more prevalent in the largest units (*Figure 3D and E*). In SOD1$^{G93A}$ mice, 40% of the FF units (6/15 tested) were unable to fire repetitively. Among the FR units, 20% (11/55 tested) were unable to fire repetitively, but all the non-firing cells were among the largest FR motor unit (1.5–4.0 mN): if we split the FR population in two along the median (1.3 mN), 46% of the large FR (11/24 tested) were unable to fire repetitively, whereas 100% of the small FR units fired repetitively (31/31 tested). Similarly, all the S units were able to fire repetitively (8/8 tested). ANOVA analysis confirmed that the motor units that did not fire repetitively developed significantly more force than those that were able to fire repetitively, regardless of the genotype (p<0.001, *Figure 3F*).

As expected, the motoneurons' input conductances were, on average, larger in FF motor units than in FRs, which were in turn larger than in slow motoneurons (one-way ANOVA, p<0.0001; *Table 2*). The motoneurons that were unable to discharge repetitively had a larger input conductance than those that still fired repetitively, regardless of the genotype (two-way ANOVA, p<0.0001; *Figure 3G*), confirming that this loss of function selectively involves the largest motoneurons that innervate the largest motor units. Interestingly, the motoneurons that did not fire repetitively had an average resting membrane potential more depolarized than those that did, regardless of their genotype (two-way ANOVA, p=0.01): non-repetitively-firing SOD1$^{G93A}$ motoneurons had an average resting membrane potential of −62 ± 8 mV (N = 17), while those that fired repetitively rested at −68 ± 8 mV (N = 63)(*Figure 3H*). Overall, we observed that the largest motor units (FF and FR larger than 1.3 mN) were more depolarized in SOD1$^{G93A}$ mice than in controls (*Table 2*), which is consistent with the fact that there is a large proportion of non-repetitively-firing motoneurons in the large units in mutant mice.

**Table 2.** Electrophysiological properties of the different types of motoneurons in WT and SOD1$^{G93A}$ mice.

Same organization as in **Table 1**. ΔV: voltage excursion between resting membrane potential and firing threshold, measured as described in the Materials and methods section. Note that ΔV, recruitment current and F-I curve slope values could only be measured in repetitively firing motoneurons.

| Property | Motor unit type | WT | SOD1$^{G93A}$ | Diff. |
|---|---|---|---|---|
| Resting membrane potential | Large units | −71 ± 9 mV; [−83−−50 mV]; N = 32 | −65 ± 9 mV; [−84−−50 mV]; N = 39 | * (p=0.013) |
| | Small units | −69 ± 10 mV; [−87−−51 mV]; N = 31 | −68 ± 7 mV; [−78−−50 mV]; N = 39 | NS |
| | FF | −73 ± 8 mV; [−83−−62 mV]; N = 6 | −67 ± 10 mV; [−84−−54 mV]; N = 15 | NS |
| | FR | −70 ± 10 mV; [−87−−50 mV]; N = 54 | −66 ± 8 mV; [−79−−50 mV]; N = 55 | * (p=0.029) |
| | S | −67 ± 4 mV; [−71−−62 mV]; N = 3 | −68 ± 9 mV; [−76−−50 mV]; N = 8 | - |
| Input conductance | Large units | 0.6 ± 0.2 μS; [0.2–0.9 μS]; N = 36 | 0.5 ± 0.2 μS; [0.2–0.8 μS]; N = 38 | NS |
| | Small units | 0.3 ± 0.1 μS; [0.1–0.4 μS]; N = 33 | 0.3 ± 0.1 μS; [0.1–0.5 μS]; N = 43 | NS |
| | FF | 0.7 ± 0.1 μS; [0.4–0.9 μS]; N = 11 | 0.6 ± 0.2 μS; [0.3–0.8 μS]; N = 15 | NS |
| | FR | 0.4 ± 0.2 μS; [0.1–0.8 μS]; N = 55 | 0.4 ± 0.1 μS; [0.1–0.8 μS]; N = 57 | NS |
| | S | 0.2 ± 0.0 μS; [0.2–0.2 μS]; N = 3 | 0.2 ± 0.1 μS; [0.1–0.3 μS]; N = 9 | - |
| ΔV | Large units | 18 ± 7 mV; [4–30 mV]; N = 19 | 21 ± 4 mV; [13–28 mV]; N = 17 | NS |
| | Small units | 17 ± 8 mV; [4–36 mV]; N = 29 | 17 ± 6 mV; [3–30 mV]; N = 37 | NS |
| | FF | [29.90 mV]; N = 1 | 22 ± 5 mV; [13–28 mV]; N = 8 | - |
| | FR | 17 ± 7 mV; [4–36 mV]; N = 45 | 18 ± 5 mV; [3–30 mV]; N = 38 | NS |
| | S | 11 ± 8 mV; [5–17 mV]; N = 2 | 16 ± 5 mV; [9–23 mV]; N = 8 | - |
| Recruitment current | Large units | 10 ± 4 nA; [2–18 nA]; N = 27 | 8 ± 3 nA; [3–12 nA]; N = 19 | NS |
| | Small units | 4 ± 3 nA; [1–17 nA]; N = 31 | 4 ± 2 nA; [0–9 nA]; N = 39 | NS |
| | FF | 13 ± 3 nA; [9–17 nA]; N = 4 | 10 ± 3 nA; [5–12 nA]; N = 7 | - |
| | FR | 7 ± 4 nA; [1–18 nA]; N = 51 | 5 ± 3 nA; [0–11 nA]; N = 43 | NS |
| | S | 2 ± 2 nA; [1–4 nA]; N = 3 | 3 ± 1 nA; [0–4 nA]; N = 8 | - |
| F-I curve slope | Large units | 11 ± 6 Hz/nA; [2–25 Hz/nA]; N = 15 | 13 ± 5 Hz/nA; [4–26 Hz/nA]; N = 17 | NS |
| | Small units | 15 ± 6 Hz/nA; [2–37 Hz/nA]; N = 28 | 19 ± 10 Hz/nA; [7–44 Hz/nA]; N = 35 | NS |
| | FF | [13.98 Hz/nA]; N = 1 | 14 ± 6 Hz/nA; [6–26 Hz/nA]; N = 6 | - |
| | FR | 14 ± 7 Hz/nA; [2–37 Hz/nA]; N = 39 | 18 ± 9 Hz/nA; [4–44 Hz/nA]; N = 38 | NS |
| | S | 13 ± 5 Hz/nA; [11–19 Hz/nA]; N = 3 | 17 ± 8 Hz/nA; [9–34 Hz/nA]; N = 8 | - |

DOI: https://doi.org/10.7554/eLife.30955.007

## The motoneurons that still discharge repetitively retain normal excitability

One measure of a motoneuron excitability is its *recruitment current*, which is the current at which it starts to fire repetitively during the ramp. The recruitment current is linearly related to the input conductance of the motoneuron: the larger the input conductance, the larger the recruitment current (*Figure 4A*, $r^2$ = 0.81, p<0.0001). Therefore, similarly to the input conductance, the recruitment current was on average larger in the FF motoneurons than in the FRs, where it was, in turn, larger than in the S motoneurons (Kruskal-Wallis rank sum test, p<0.0001; *Table 2*). Compared with WT mice, the recruitment current was unchanged for all motoneuron types in SOD1$^{G93A}$ mice compared with the WT mice (*Table 2*). This holds true when all the motoneuron types are pooled together (WT: 7.0 ± 4.6 nA; N = 58 vs. SOD1$^{G93A}$: 5.4 ± 3.0 nA; N = 60; Mann-Whitney p=0.14; *Figure 4B*). Comparing WT and SOD1$^{G93A}$ motoneurons, we found that the slope of the relationship between the recruitment current and the input conductance was not significantly different (ANCOVA p=0.99, *Figure 4A*), indicating that the excitability of the motoneurons that retain the ability to fire repetitively was unaffected by the mutation. The voltage excursion (difference between the spiking voltage threshold and the resting membrane potential) was also unaffected in SOD1$^{G93A}$ vs. WT motoneurons (WT: 17.4 ± 7.3 mV; N = 48 vs. SOD1$^{G93A}$: 18.2 ± 5.3 mV; N = 56; Mann-Whitney p=0.64;

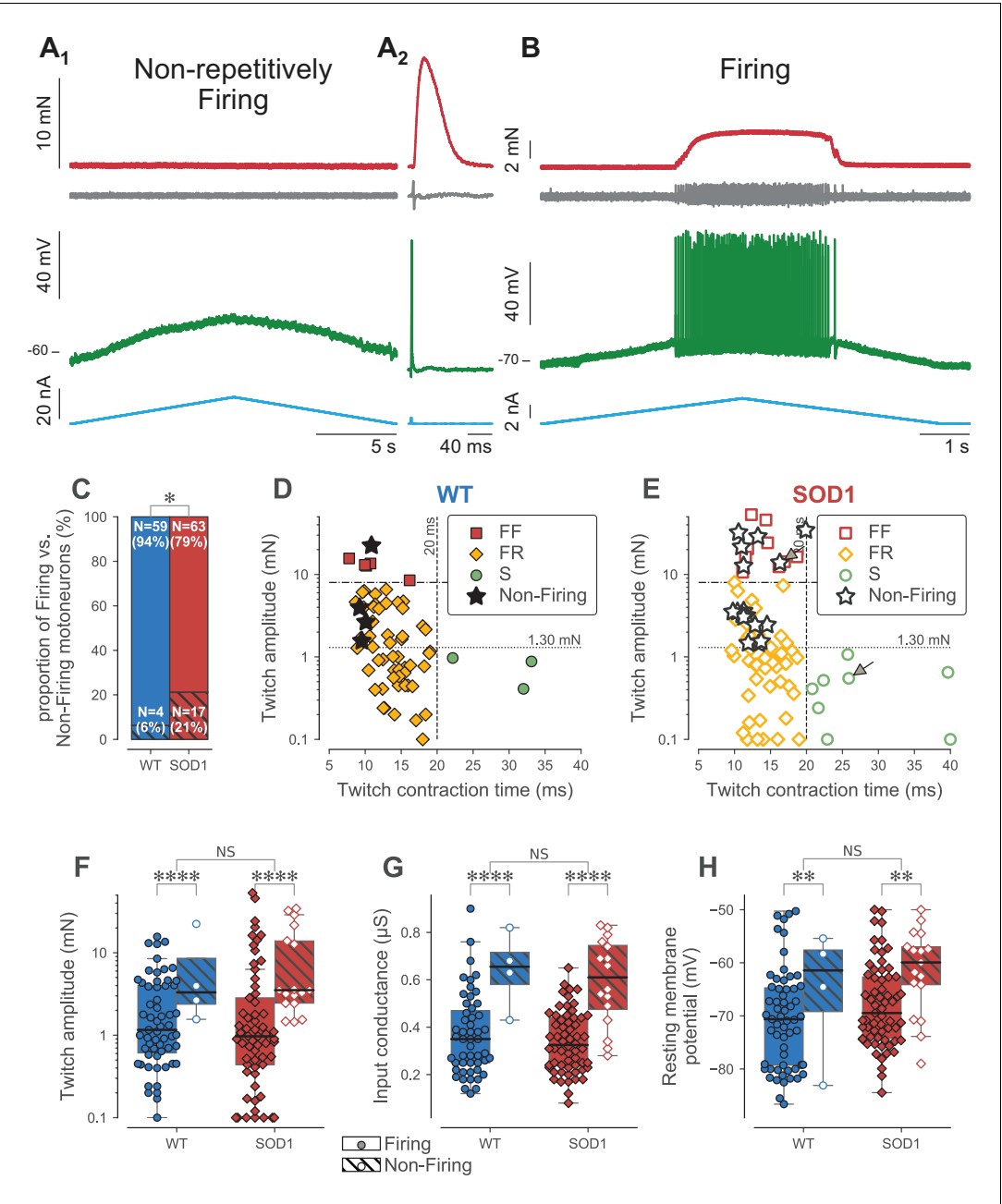

**Figure 3.** Loss of repetitive firing in a subpopulation of large motor units. (A) Example of an FF-type SOD1[G93A] motoneuron that was unable to fire repetitively in response to a slow ramp of current (A₁), despite being able to generate a single spike in response to a short pulse of current (A₂), and despite being still connected to its muscle fibers, as shown by the presence of a motor unit action potential and a motor unit twitch following the spike. Top red trace: force developed by the motor unit. Grey trace, second from the top: EMG recording showing the motor unit action potentials. Green trace, second from bottom: membrane potential. Bottom blue trace: injected current. A₂ is an average of 10 sweeps. (B) Example of a S-type SOD1[G93A] motoneuron that was able to fire repetitively in response to a slow ramp of current. Same organization as in A. (C) Comparison of the proportion of Firing (filled bar) and Non-Firing (hatched bar) motoneurons in WT (blue) vs. SOD1[G93A] mice (red). (D) Contractile properties of WT motor units in which we tested the ability to fire repetitively. The motoneurons that were unable to fire repetitively are indicated by a star. The dashed lines at 8 mN and 20 ms represent the limits used to classify the motor units, and the dotted line and 1.3 mN represent the separation line between Large and Small motor units. (E) Contractile properties of SOD1[G93A] motoneurons in which we tested the ability to fire repetitively. Motoneurons indicated by an arrow correspond to the two examples in panels A and B. Same legend as in D.

*Figure 3 continued on next page*

*Figure 3 continued*

Comparison of the twitch amplitude (**F**), input conductance (**G**) and resting membrane potential (**H**) of WT (blue circles) and SOD1$^{G93A}$ (red diamonds) motoneurons based on whether they were able (filled symbols, empty box-and-whisker plot) or unable (empty symbols, hatched box-and-whisker plot) to fire, repetitively. The box-and-whisker plots are defined as follows: the boxes extend from the first to third quartile values of the data, with a line at the median. The whiskers extend from the box up to 1.5 times the interquartile range to show the range of the data.

DOI: https://doi.org/10.7554/eLife.30955.005

*Figure 4C*; and *Table 2* for the distributions among motoneuron types). As a final measure of excitability, we looked at motoneuron gain. When the motoneuron reaches its recruitment current, the discharge frequency increases with the injected current (first within the sub-primary range where the discharge is irregular and increases non-linearly and then within the primary range where the discharge is regular and increases linearly; *Manuel et al., 2009*). The gain is defined as the slope of the frequency-current relationship in the primary range. In the whole population of motoneurons that fired during the ramp, we found that the gain was unchanged (WT: 14 ± 6 Hz/nA; N = 43 vs. SOD1$^{G93A}$: 17 ± 9 Hz/nA; N = 54 Mann-Whitney p=0.10; *Figure 4D*), and also in the motoneuron subtypes taken individually (*Table 2*). Overall, our results show that the motoneurons that become hypoexcitable by losing the ability to fire repetitively all belong to the FF motor units or the largest FR motor units whereas the remaining SOD1$^{G93A}$ motoneurons retain their normal excitability.

## Similar loss of function in FUS$^{P525L}$ transgenic mice

In order to test if the loss of function observed in SOD1$^{G93A}$ mice is specific to this particular mouse model of ALS, we studied the electrophysiological properties of spinal motoneurons in an unrelated model of familial ALS based on the overexpression of mutant human FUS (*Sharma et al., 2016*). In contrast to the SOD1$^{G93A}$ model, where denervation occurs fairly synchronously across hindlimb motor pools (*Pun et al., 2006*), denervation of Tibialis Anterior (TA, an ankle flexor muscle) in the $\tau^{ON}$FUS$^{P525L}$ mouse precedes denervation of TS (an ankle extensor)(*Sharma et al., 2016*). Moreover, the progression of the denervation is much slower in $\tau^{ON}$FUS$^{P525L}$ than in SOD1$^{G93A}$ mice. We therefore studied the electrical properties of type-identified ankle flexor motor units or ankle extensors at two time points: P30 (at which time the denervation process has just started in TA), and P180 (when moderate amount of denervation is observed in TS), and compared them to control mice in which WT human FUS is expressed from the same locus ($\tau^{ON}$FUS$^{WT}$). Despite some small variations in twitch amplitudes, the TS motor unit properties were remarkably similar to those recorded in the SOD1$^{G93A}$ mouse, regardless of the age (compare *Figure 2A–B* and *Figure 5C–D*). Consistent with the fact that TA and EDL are almost devoid of Type I muscle fibers (*Wang and Kernell, 2001*; *Bloemberg and Quadrilatero, 2012*), motor units recorded from the ankle flexors were all fast-contracting, but otherwise displayed similar twitch amplitudes as in the ankle extensors (*Figure 5E–F*).

As in SOD1$^{G93A}$ mice, we observed a large proportion of motoneurons in $\tau^{ON}$FUS$^{P525L}$ mice that did not fire repetitively in response to a slow ramp of current (*Figure 5A$_1$*), while others were able to fire normally (*Figure 5B*). Again, this loss of function occurred at a time when the neuromuscular junctions are not denervated, as shown by the fact that an action potential elicited by a transient electrical pulse in the cell body elicits a motor unit action potential and a twitch of the motor unit (*Figure 5A$_2$*). However, the proportion of motoneurons incapable of repetitive firing depended on motor pool and age (*Figure 5G–H*). In the ankle extensor motor pools, where denervation is the slowest, the proportion of non-firing motoneurons was not statistically different between WT and mutant animals at either time point studied (Fisher exact test, p=0.69 at P30, p=1.00 at P180). In the ankle flexor motor pools, the proportion of non-firing motoneurons was not statistically significant at P30, when the denervation is just starting (Fisher exact test, p=0.43). However, at P180, when the denervation is more advanced, there were significantly more non-repetitively-firing motoneurons in FUS$^{P525L}$ than in FUS$^{WT}$ mice (42% vs. 18%, respectively; Fisher exact test, p=0.03). This occurs despite of the fact that the contractile properties of the ankle flexors motor units are similar between FUS$^{P525L}$ and FUS$^{WT}$ mice (*Table 3*). As before, this loss of function did not affect all motor unit types equally, but was strongly biased toward the largest motor units. In FF units, we found that 7/9 (78%) motoneurons were unable to fire repetitively (vs. 2/7, 17% in WT mice; Fisher exact test p=0.04). In

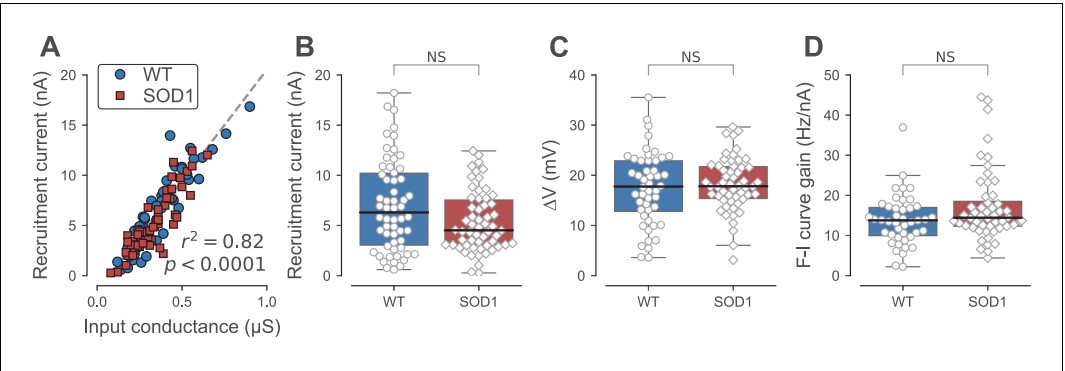

**Figure 4.** Firing properties of WT and SOD1$^{G93A}$ motoneurons. (**A**) Relationship between recruitment current and input conductance of WT (blue circles) vs. SOD1$^{G93A}$ (red diamonds) motoneurons. (**B**) Comparison of the current required to elicit the first spike on a ramp of current (recruitment current) of WT (blue box, circles) vs. SOD1$^{G93A}$ (red box, diamonds) motoneurons. (**C**) Comparison of the distance between the resting membrane potential and the voltage threshold for spiking ($\Delta V$) of WT (blue box, circles) vs. SOD1$^{G93A}$ (red box, diamonds) motoneurons. (**D**) Comparison of the F-I curve gains (slope of the frequency versus injected current curve measured in the primary range) of WT (blue box, circles) vs. SOD1$^{G93A}$ (red box, diamonds) motoneurons. In all panels, the definition of the box-and-whisker plots is the same as in *Figure 3*.

DOI: https://doi.org/10.7554/eLife.30955.008

FR units (large and small taken together), we found that only 7/22 (32%) motoneurons were unable to fire repetitively, which was not statistically different from WT mice (4/21, 19%; Fisher exact test p=0.49). As we did with SOD1 mice, when we split our samples between 'large' and 'small' motor units (based on the same cutoff value of 1.3 mN as used above, which is also the median twitch amplitude of FR ankle flexor units), the proportion of non-firing motoneuron was much larger in the large ankle flexor motor units in FUS$^{P525L}$ vs. FUS$^{WT}$ mice at P180 (13/19, 68% vs. 3/17, 18%, respectively; Fisher exact test, p=0.003) whereas the proportion was not statistacally different in the small motor units (WT: 2/10, 20% vs. FUS: 1/12, 8%; Fisher exact test, p=0.57).

*Table 4* shows that the electrophysiological properties of 180 days-old ankle flexor motoneurons were not different between FUS$^{WT}$ and FUS$^{P525L}$ in each physiological type. However, and strikingly similar to the results of the SOD1 model, we observed that the resting membrane potential of ankle flexor motoneurons at P180 tended to be more depolarized in non-firing cells than in cells able to fire repetitively, regardless of the phenotype (FUS$^{WT}$: non-firing $-60 \pm 7$ mV; N = 6 vs. firing $-72 \pm 6$ mV; N = 28; FUS$^{P525L}$: non-firing $-65 \pm 7$ mV; N = 15 vs. firing $-70 \pm 10$ mV; N = 21; two-way ANOVA, p=0.001; *Figure 6A*). Also in keeping with our results in the SOD1 model, the ankle flexor motoneurons that were able to fire repetitively at P180 retained normal excitability. The recruitment current of the firing motoneurons was not significantly different from controls (FUS$^{P525L}$: 7 $\pm$ 5 nA; N = 21 vs. FUS$^{WT}$: 6 $\pm$ 4 nA; N = 28; Mann-Whitney p=0.87; *Figure 6B*). Moreover, the slope of the relationship between recruitment current and input conductance was not affected by the mutation (ANCOVA p=0.19; *Figure 6C*).

## Motoneuron chondrolectin expression follows the pattern of firing abnormality

Altogether, our results indicate that the loss of function is largely confined to the FF and the largest FR motoneurons, which we identified on the basis of their physiological properties. We were also interested to find a molecular marker of these selectively vulnerable motoneurons. Using in situ hybridization, we looked at the pattern of expression of the C-type lectin chondrolectin (*Chodl*), which has been implicated in motor axon growth during development (*Zhong et al., 2012*; *Sleigh et al., 2014*). *Chodl* was suggested to be selectively expressed in F-type motoneurons (*Enjin et al., 2010*) and it has been shown to disappear during the disease progression in the SOD1$^{G93A}$ mice, suggesting that it is somehow linked to motoneuron vulnerability (*Wootz et al., 2010*). In 19 WT TS motoneurons that we labeled intracellularly with neurobiotin during electrophysiological experiments (only one motoneuron was labeled per experiment)(*Figure 7A*), we found that

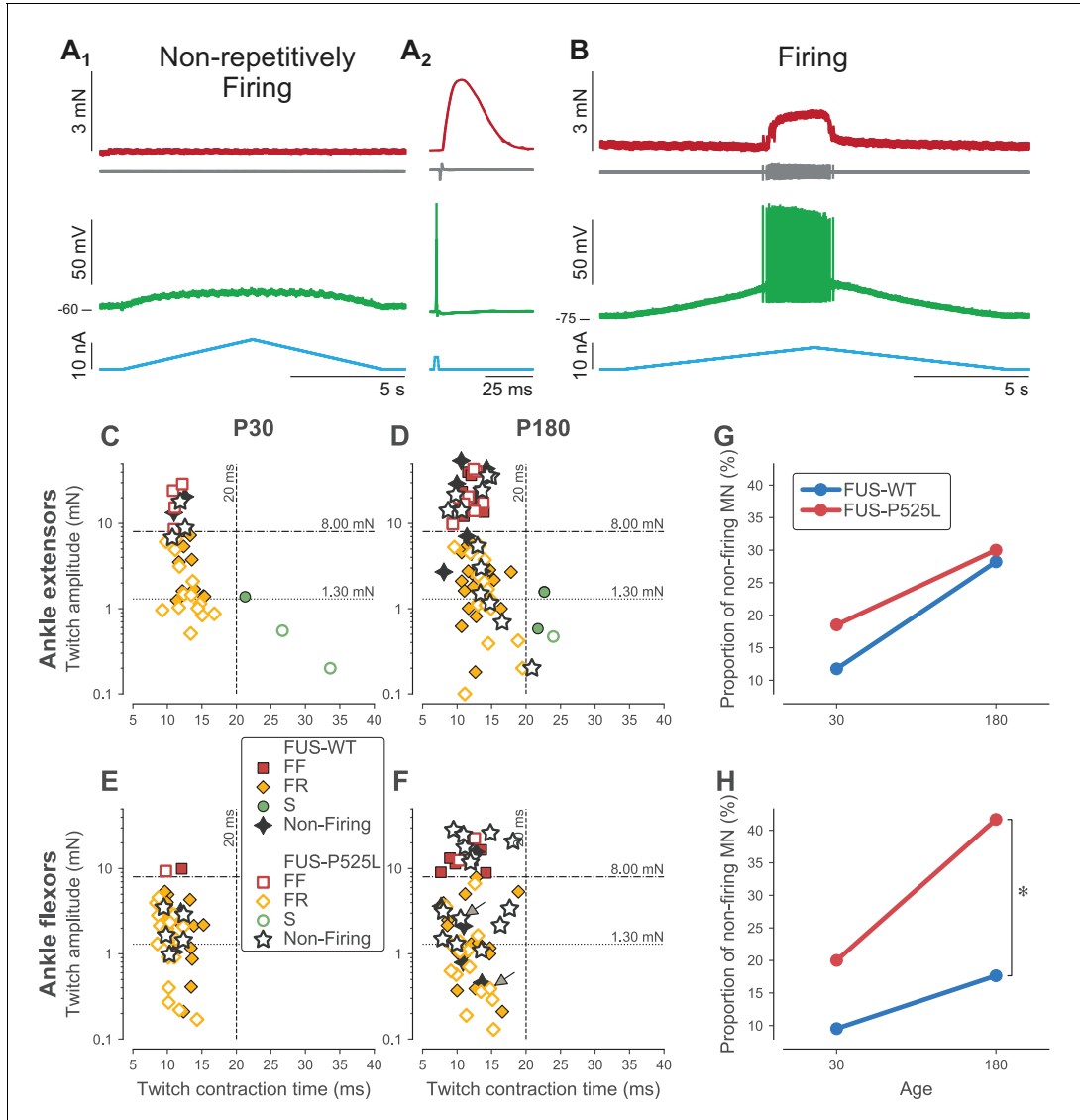

**Figure 5.** Loss of repetitive firing in a subpopulation of cells of FUS mice. (**A**) Example of an ankle flexor FUS[P525L] motoneuron (recorded at P180) that was unable to fire repetitively in response to a slow ramp of current (**A₁**), despite being able to generate a single spike in response to a short pulse of current (**A₂**), and despite being still connected to its muscle fibers, as shown by the presence of a motor unit action potential and a motor unit twitch following the spike. Top red trace: force developed by the motor unit. Grey trace, second from the top: EMG recording showing the motor unit action potentials. Green trace, second from bottom: membrane potential. Bottom blue trace: injected current. A₂ is an average of 15 sweeps. (**B**) Example of an ankle flexor FUS[P525L] motoneuron (recorded at P180) that was able to fire repetitively in response to a slow ramp of current. Same organization as in A. (**C**) Contractile properties of motor units innervating ankle extensor muscles at P30. Filled symbols represent motor units recorded in FUS[WT] mice, colored according to their physiological type (FF: red squares; FR: yellow diamonds, S: green circles). WT motor units that were unable to fire repetitively are represented with a black four-pointed star. Empty symbols represent motor units recorded in FUS[P525L], colored according to their physiological type (FF: red squares; FR: yellow diamonds, S: green circles). Mutant motor units that were unable to fire repetitively are represented with a black empty five-pointed star. The dashed lines at 8 mN and 20 ms represent the limits used to classify the motor units, and the dotted line and 1.3 mN represent the separation line between Large and Small motor units. (**D**) Contractile properties of motor units innervating ankle extensor muscles at P180. Same legend as in C. (**E**) Contractile properties of motor units innervating ankle flexor muscles at P30. Same legend as in C. (**F**) Contractile properties of motor units innervating ankle flexor muscles at P180. Same legend as in C. Motoneurons indicated by an arrow correspond to the two examples in panels A and B. (**G**) Comparison between FUS[WT] (blue) and FUS[P525L] (red) mice of the proportion of non-firing cells innervating ankle extensor muscles at P30 and P180. (**H**) Comparison between FUS[WT] (blue) and FUS[P525L] (red) mice of the proportion of non-firing cells innervating ankle flexor muscles at P30 and P180.

DOI: https://doi.org/10.7554/eLife.30955.009

**Table 3.** Contractile properties of the different types of ankle flexor motor units in the in 180 days-old FUS^WT and FUS^P525L mice. Same organization as in *Table 1*.

| Property | Motor unit type | FUS^WT | FUS^P525L | Diff. |
|---|---|---|---|---|
| Twitch amplitude | Large units | 6.8 ± 5.9 mN; [1.3–22.0 mN]; N = 28 | 10.6 ± 9.7 mN; [1.3–28.4 mN]; N = 22 | NS |
| | Small units | 0.8 ± 0.3 mN; [0.2–1.3 mN]; N = 18 | 0.7 ± 0.3 mN; [0.1–1.3 mN]; N = 16 | NS |
| | FF | 14.7 ± 5.1 mN; [8.9–22.0 mN]; N = 8 | 20.0 ± 6.0 mN; [11.8–28.4 mN]; N = 10 | NS |
| | FR | 2.3 ± 2.0 mN; [0.2–8.0 mN]; N = 38 | 1.5 ± 1.4 mN; [0.1–6.7 mN]; N = 28 | NS |
| | S | - | - | - |
| Twitch contraction time | Large units | 11.0 ± 2.6 ms; [7.7–18.9 ms]; N = 28 | 11.6 ± 2.9 ms; [7.5–18.1 ms]; N = 22 | NS |
| | Small units | 12.4 ± 2.4 ms; [8.0–16.6 ms]; N = 18 | 11.4 ± 2.3 ms; [9.1–15.3 ms]; N = 16 | NS |
| | FF | 10.9 ± 2.4 ms; [7.7–14.2 ms]; N = 8 | 12.2 ± 2.5 ms; [9.5–18.1 ms]; N = 10 | NS |
| | FR | 11.7 ± 2.7 ms; [7.7–18.9 ms]; N = 38 | 11.3 ± 2.7 ms; [7.5–17.6 ms]; N = 28 | NS |
| | S | - | - | - |
| Fatigue Index | Large units | 0.9 ± 0.1; [0.8–1.0]; N = 5 | 0.8 ± 0.2; [0.4–1.0]; N = 6 | - |
| | Small units | 1.0 ± 0.1; [0.9–1.0]; N = 10 | 1.0 ± 0.0; [0.9–1.0]; N = 12 | NS |
| | FF | 0.9 ± 0.1; [0.9–1.0]; N = 2 | 0.7 ± 0.3; [0.4–0.9]; N = 2 | - |
| | FR | 0.9 ± 0.1; [0.8–1.0]; N = 13 | 0.9 ± 0.1; [0.7–1.0]; N = 16 | NS |
| | S | - | - | - |

DOI: https://doi.org/10.7554/eLife.30955.010

*Chodl* was expressed in 8 out of 9 motoneurons innervating large motor units (twitch force larger than 1.3 mN, the median value of FR units, as used above)(*Figure 7B and D*). Only 2 out of 10 motoneurons innervating smaller motor units expressed *Chodl*. On average, *Chodl*+ motor units developed 8.1 ± 8.6 mN of twitch force (range 1.0–29.8 mN; N = 10), while *Chodl*− motor units only developed 0.8 ± 0.6 mN of twitch force (range 0.4–2.2 mN; N = 9; Mann-Whitney p<0.0001; *Figure 7C*). This shows that *Chodl* is expressed only in the FF motoneurons and in the FR motoneurons that innervate the largest motor units. Altogether, these data indicate that the FR motor units are not a homogeneous population, but rather can be separated into two subpopulations according to their size. The motoneurons innervating the largest FR motor units express *Chodl* in WT mice and they are more prone to lose their response to sustained inputs in ALS mice than those innervating the smallest FR motor units.

## The loss of electrophysiological function represents a more advanced stage of disease progression

Our electrophysiological experiments together with the chondrolectin studies suggest a functional and molecular relationship between the FF and the largest FR motoneurons. However, within this population, only a fraction of motoneurons have lost the ability to fire repetitively at the time points we considered (P50 for the ankle extensors in SOD1^G93A mice and P180 for the ankle flexors in FUS^P525L mice). To test whether the non-repetitively-firing motoneurons are at a more advanced stage of disease progression, we set out to correlate firing properties with biochemical markers of the ongoing pathogenic process.

Although a number of abnormalities have been described in vulnerable motoneurons in ALS (e.g. ER stress, disruption of ER chaperones, unfolded protein response activation, and ubiquitinated proteins accumulation; *Saxena et al., 2009*; *Bernard-Marissal et al., 2012*; *Saxena et al., 2013*), how these pathways relate to the disruption of core physiological properties of motoneurons remains to be investigated. For this purpose, we selected p62 and phosphorylated-eIF2α (p-eIF2α) as disease markers. p62 is an autophagy-substrate adapter involved in the degradation of mitochondria and aggregated misfolded proteins (*Matsumoto et al., 2015*; *Demishtein et al., 2017*). In the spinal cords of ALS mouse models, p62 accumulates over time in ALS mouse models spinal cord (*Zhang et al., 2011*) and p62-positive inclusions have been reported in vulnerable motoneurons (*Kaplan et al., 2014*; *Rudnick et al., 2017*). In the case of p-eIF2α, levels first increase in the most

**Table 4.** Electrophysiological properties of the different types of ankle flexor motoneurons in the in 180 days-old FUS$^{WT}$ and FUS$^{P525L}$ mice.

Same organization as in **Table 2**.

| Property | Motor unit type | FUS$^{WT}$ | FUS$^{P525L}$ | Diff. |
|---|---|---|---|---|
| Resting membrane potential | Large units | −71 ± 9 mV; [−92–−58 mV]; N = 19 | −67 ± 9 mV; [−82–−50 mV]; N = 19 | NS |
| | Small units | −67 ± 8 mV; [−76–−51 mV]; N = 10 | −69 ± 10 mV; [−86–−54 mV]; N = 12 | NS |
| | FF | −75 ± 11 mV; [−92–−61 mV]; N = 6 | −66 ± 7 mV; [−77–−52 mV]; N = 9 | NS |
| | FR | −68 ± 7 mV; [−83–−51 mV]; N = 23 | −69 ± 10 mV; [−86–−50 mV]; N = 22 | NS |
| | S | - | - | - |
| Input conductance | Large units | 0.5 ± 0.2 µS; [0.2–1.1 µS]; N = 19 | 0.5 ± 0.2 µS; [0.2–0.8 µS]; N = 17 | NS |
| | Small units | 0.2 ± 0.1 µS; [0.1–0.4 µS]; N = 10 | 0.3 ± 0.1 µS; [0.1–0.5 µS]; N = 11 | NS |
| | FF | 0.6 ± 0.2 µS; [0.3–1.1 µS]; N = 8 | 0.5 ± 0.2 µS; [0.2–0.8 µS]; N = 9 | NS |
| | FR | 0.3 ± 0.2 µS; [0.1–0.8 µS]; N = 21 | 0.3 ± 0.1 µS; [0.1–0.7 µS]; N = 19 | NS |
| | S | - | - | - |
| ΔV | Large units | 20 ± 7 mV; [12–34 mV]; N = 13 | 24 ± 5 mV; [20–33 mV]; N = 6 | NS |
| | Small units | 15 ± 6 mV; [10–26 mV]; N = 9 | 16 ± 6 mV; [8–24 mV]; N = 9 | NS |
| | FF | 24 ± 7 mV; [15–34 mV]; N = 5 | 27 ± 9 mV; [21–33 mV]; N = 2 | - |
| | FR | 17 ± 6 mV; [10–30 mV]; N = 17 | 18 ± 6 mV; [8–27 mV]; N = 13 | NS |
| | S | - | - | - |
| Recruitment current | Large units | 9 ± 3 nA; [4–14 nA]; N = 14 | 10 ± 4 nA; [5–15 nA]; N = 6 | NS |
| | Small units | 4 ± 2 nA; [1–7 nA]; N = 8 | 6 ± 5 nA; [1–20 nA]; N = 11 | NS |
| | FF | 10 ± 3 nA; [7–14 nA]; N = 5 | 14 ± 2 nA; [12–15 nA]; N = 2 | - |
| | FR | 6 ± 3 nA; [1–12 nA]; N = 17 | 7 ± 5 nA; [1–20 nA]; N = 15 | NS |
| | S | - | - | - |
| F-I curve slope | Large units | 9 ± 5 Hz/nA; [4–21 Hz/nA]; N = 11 | 11 ± 5 Hz/nA; [5–17 Hz/nA]; N = 5 | - |
| | Small units | 29 ± 13 Hz/nA; [7–48 Hz/nA]; N = 7 | 17 ± 15 Hz/nA; [4–43 Hz/nA]; N = 9 | NS |
| | FF | 6 ± 1 Hz/nA; [5–7 Hz/nA]; N = 3 | [14.18 Hz/nA]; N = 1 | - |
| | FR | 19 ± 14 Hz/nA; [4–48 Hz/nA]; N = 15 | 15 ± 13 Hz/nA; [4–43 Hz/nA]; N = 13 | NS |
| | S | - | - | - |

DOI: https://doi.org/10.7554/eLife.30955.011

vulnerable motoneurons soon after the peak of ER stress markers is reached (about P37–42) and then decline back to baseline; p-eIF2α upregulation then follows in less vulnerable motoneurons (**Saxena et al., 2009**).

To correlate disease markers to electrophysiology, recorded cells from SOD1$^{G93A}$ mice aged 50 ± 2 days old (N = 6 mice) were labeled with neurobiotin, then immunostained for p-eIF2α and p62 (**Figure 8**). Altogether, we recorded eight motoneurons (four firing repetitively and four unable to do so). To maximize in vivo intracellular labeling of motoneurons, mice were treated with curare, which prevented us from identifying their physiological type. However, the recorded motoneurons were large cells and most likely belonged to large-sized motor units as they were large cells. The soma cross-sectional area of the whole population of α-motoneurons in the imaged slices ranged from 80 to 1030 µm$^2$ (420 ± 193 µm$^2$; N = 279), while all the labeled motoneurons were larger than 490 µm$^2$ (and up to 900 µm$^2$; 704 ± 167 µm$^2$; N = 8). In addition, their input conductances (0.5 ± 0.1 µS; N = 8) were in the expected range for large units (**Table 2**). As previously, the non-repetitively-firing motoneurons had larger input conductances than the repetitively firing motoneurons (firing: 0.4 ± 0.1 µS; N = 4 vs. non-firing: 0.6 ± 0.1 µS; N = 4; Mann-Whitney p=0.030).

As expected, p-eIF2α was expressed in all NB-labeled motoneurons, while small, resistant size motoneurons were devoid of p-eIF2α (**Figure 8-Figure supplement 1**), indicating that all the NB-labeled motoneurons have experienced ER-stress. However, the levels of p-eIF2α were significantly

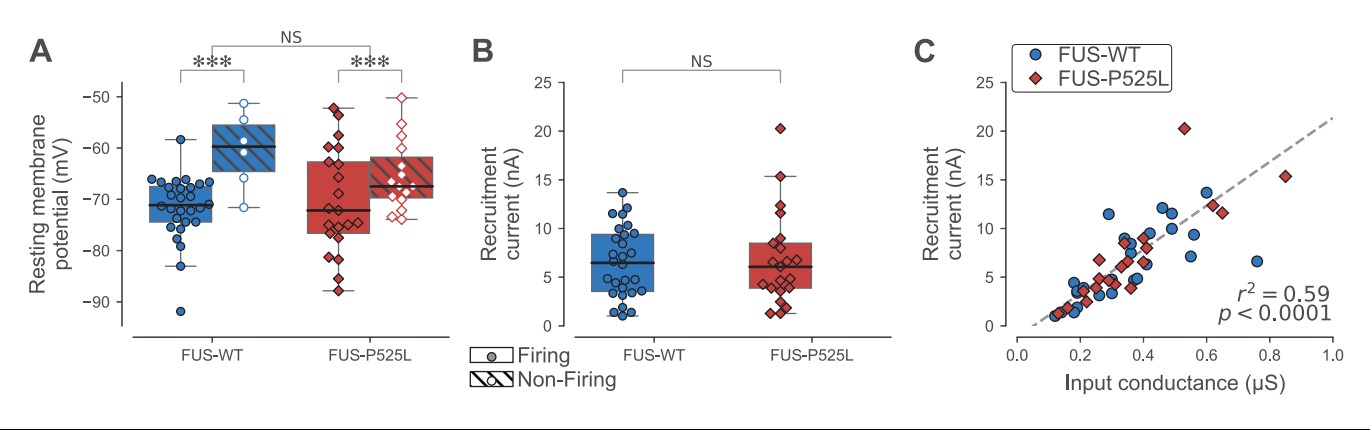

**Figure 6.** Electrical properties of FUS[WT] and FUS[P525L] motoneurons innervating ankle flexor muscles at P180. (**A**) Comparison of the resting membrane potential of FUS[WT] (blue box, circles) vs. FUS[P525L] (red box, diamonds) motoneurons innervating ankle flexor muscles at P180, split according to whether they were able (filled symbols) to fire repetitively or not (empty symbols, hatched box-and-whisker plot). (**B**) Comparison of the current required to elicit the first spike on a ramp of current (recruitment current) of FUS[WT] (blue box, circles) vs. FUS[P525L] (red box, diamonds) motoneurons. (**C**) Relationship between recruitment current and input conductance of FUS[WT] (blue circles) vs. FUS[P525L] (red diamonds) motoneurons. In all panels, the definition of the box-and-whisker plots is the same as in *Figure 3*.

DOI: https://doi.org/10.7554/eLife.30955.012

higher in repetitively firing motoneurons than in the motoneurons that were unable to fire repetitively (*Figure 8A₃ and B₃*; firing: 0.8 ± 0.2; N = 4 vs. non-firing: 0.2 ± 0.2; N = 4; Mann-Whitney p=0.02; *Figure 8C*). Given that p-eIF2α levels decline in the most vulnerable motoneurons after P42 and rise in the more resistant motoneurons at about P50 (*Saxena et al., 2009*), our results suggest that the non-repetitively-firing motoneurons are the most vulnerable and have started to downregulate p-eIF2α, while less vulnerable motoneurons, which discharge repetitively, are still in the early stages of the biochemical progression of the disease.

All the NB-labeled motoneurons displayed p62 aggregates, in contrast to smaller neighboring motoneurons, which did not (*Figure 8-Figure supplement 1*), indicating again that all the investigated motoneurons were vulnerable. However, in the four repetitively-firing NB-labeled motoneurons (from three different mice), p62 burden was moderate compared to some neighboring motoneurons that exhibited higher amounts of aggregates (*figure 8A₂*). In the non-repetitively-firing motoneurons, p62 expression was more variable; two neurons, recorded in two different mice, displayed a p62 burden as high as in any of the neighboring motoneurons (*Figure 8B₂*), indicating that these two non-firing motoneurons were in an advanced stage of the disease. However, in the other two non-repetitively-firing motoneurons, both recorded in the same mouse, p62 burden was lower.

To summarize, by the time p-eIF2α expression started to increase in the repetitively firing motoneurons, p-eIF2α was already downregulated in the non-firing cells, indicating that the non-firing motoneurons are more vulnerable (*Saxena et al., 2009*). Moreover, whereas p62 burden was moderate in firing motoneurons, it was high in two of the non-repetitively-firing motoneurons. Altogether, our results strongly suggest that the non-firing, hypoexcitable, state corresponds to a more advanced stage of the disease than the repetitive firing state.

## Discussion

For the first time in a mouse model of ALS, we have been able to correlate changes in the electrical properties of motoneurons with the physiological type of their motor unit. The identification of the physiological type was based on the measurement of the force developed by the investigated motor unit, while simultaneously recording the membrane potential of its motoneuron. This allowed us to demonstrate that, in two unrelated mouse models of the disease, the motoneurons that lose their ability to fire repetitively in response to sustained input were distributed among the motoneurons that innervate the FF motor units and the largest FR motor units. A molecular marker, *Chodl*, provides further evidence that the FR motor units can be divided into two-subpopulations depending

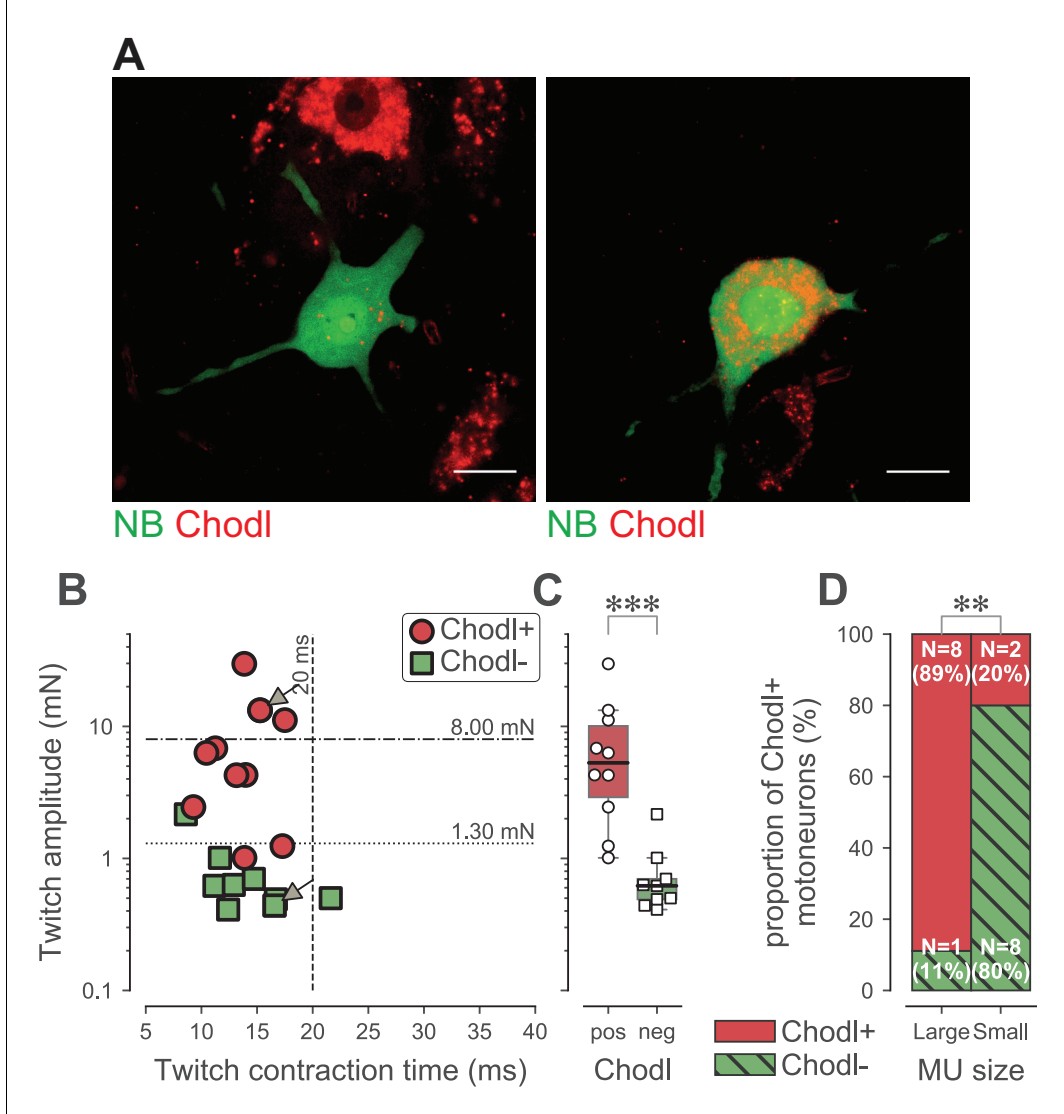

**Figure 7.** Chondrolectin expression in TS motor units. (**A**) Two examples of intracellular-labeled (NB, green) motoneurons, coupled with ISH revelation of Chodl RNA (Chodl, red). Left panel: Chodl− small FR motoneuron; Right panel: Chodl +FF motoneuron. Scale bars: 15 μm. (**B**) Contractile properties of the motor units tested for chondrolectin expression. The motoneurons indicated with arrows correspond to the two cells in A. Red circles are the motoneurons that expressed chondrolectin, while green squares are those that did not. The dashed lines at 8 mN and 20 ms represent the limits used to classify the motor units, and the dotted line at 1.3 mN represent the separation line between Large and Small motor units. (**C**) Comparison of the average twitch amplitude of motor units split according to their expression of chondrolectin. Same legend as in B. The definition of the box-and-whisker plots is the same as in *Figure 3*. (**D**) Comparison of the proportion of cells expressing chondrolectin in the population of tested cells, split in two categories, large and small, according to a limit set to 1.3 mN.
DOI: https://doi.org/10.7554/eLife.30955.013

on their size. The smallest FR motor units do not express *Chodl,* whereas the largest FR motor units express *Chodl* along with the FF motor units. Furthermore, our data indicate that the motoneurons that failed to discharge repetitively are farther along in the neurodegenerative process than those that retain the ability to fire repetitively, as suggested by the differential expression of p-eIF2α and p62 aggregates. This study demonstrates that motoneuron hypoexcitability—as measured by the inability to fire repetitively—is a loss of function that precedes the withdrawal of motor axons from the neuromuscular junction, and is likely to be an early sign of motoneuron degeneration in ALS.

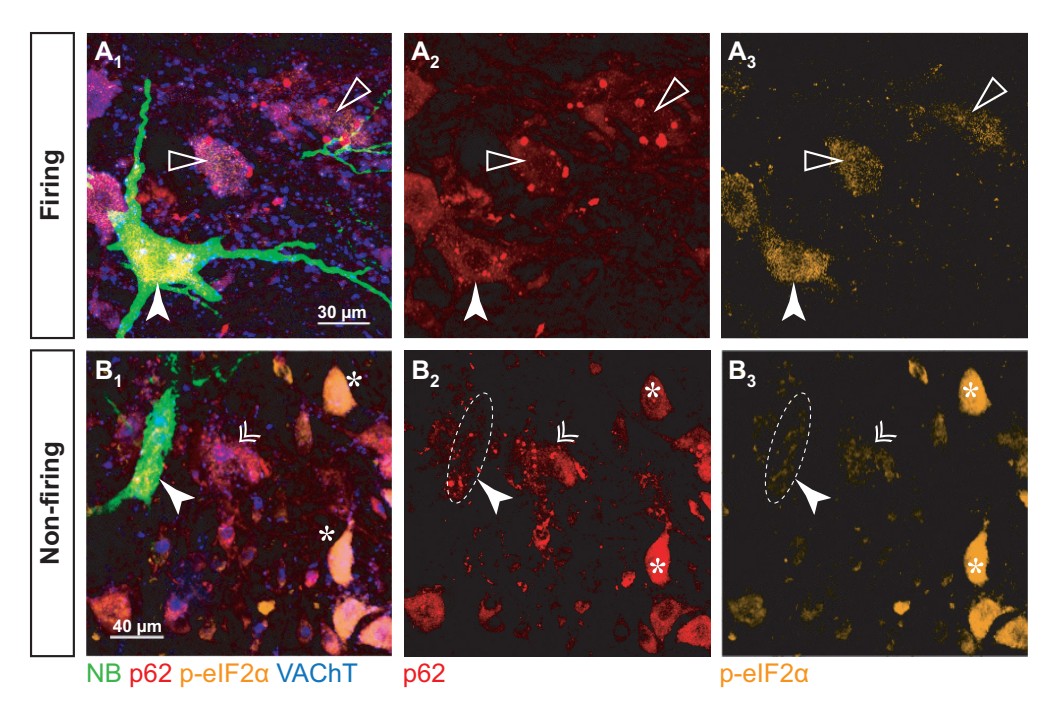

**Figure 8.** p-eIF2α and p62 burden in firing and non-firing SOD1[G93A] motoneurons. (**A**) Example of a neurobiotin-labeled repetitively firing motoneuron. (**A₁**) shows the overlay of the neurobiotin (green), p62 (red), p-eIF2α (orange) and VAChT (blue). (**A₂**) and **A₃** show the p62 and p-eIF2α labeling, respectively. The labeled motoneuron is indicated with a filled arrowhead. Empty arrowheads point to neighboring motoneurons with higher p62 burden but lower p-eIF2α fluorescence. (**B**) Example of a neurobiotin-labeled non-repetitively-firing motoneuron. Same organization as in A. The labeled motoneuron is indicated with a filled arrowhead. The double arrow point to a neighboring motoneuron with a similar p62 and p-eIF2α burden. The asterisks indicate other motoneurons with high p-eIF2α labeling but no p62 aggregates.

DOI: https://doi.org/10.7554/eLife.30955.014

The following figure supplement is available for figure 8:

**Figure supplement 1.** Small motoneurons express low levels of disease markers.

DOI: https://doi.org/10.7554/eLife.30955.015

## Correlation between electrical behavior of motoneurons and contractile properties of their motor units

The first in vivo intracellular recordings of spinal motoneurons in mice made it possible to investigate the intrinsic electrical properties of motoneurons (*Huizar et al., 1975*; *Alstermark and Ogawa, 2004*; *Manuel et al., 2009*; *Meehan et al., 2010*; *Nakanishi and Whelan, 2012*). However, for the sake of mechanical stability of the intracellular recordings, the mouse was paralyzed in these early studies, which made it impossible to record the contractile properties of the innervated muscle fibers. Subsequent technical improvements opened the door to stable recordings in anesthetized and non-paralyzed mice (*Manuel and Heckman, 2011*), making it possible to simultaneously record the electrophysiological properties of single motoneurons and the contractile force of the innervated muscle fibers, that is their specific motor unit (*Manuel and Heckman, 2011*). In the present study, we used this technique to investigate motor units in which the neuromuscular junctions were still functioning, which allowed us to determine the force profile of individual motor units. We first identified the physiological type of recorded motor units based on their twitch amplitude, contraction velocity and resistance to fatigue, and then correlated the electrical behavior of motoneurons with the physiological type of their motor units. This was done on both control and ALS mice. Using this direct approach, we demonstrated that the motoneurons that lose their ability to discharge repetitively in ALS mice (*Delestrée et al., 2014*) are distributed among the FF motor units, which are the most vulnerable motor units in ALS (*Pun et al., 2006*; *Hegedus et al., 2008*), but also among the FR

motor units with a twitch amplitude larger than 1.3 mN. Furthermore, using *Chodl* as a marker of neurodegeneration in ALS (*Wootz et al., 2010*), we examined the distribution pattern of *Chodl* in a number of type-identified motoneurons in WT mice. By coupling in vivo electrophysiology (motor unit type-identification and motoneuron intracellularly labeling) and post-hoc in situ hybridization, we found that *Chodl* is mainly expressed in motor units that develop a twitch force larger than 1.3mN, demonstrating that these motor units that lose the ability to fire repetitively are selectively vulnerable in ALS. As further evidence that specific physiological subtypes of motoneurons are selectively vulnerable in ALS, we found that *Chodl* is also selectively expressed in the largest FR motor units, which tend to lose their ability to fire rhythmically in response to sustained inputs.

## Loss of function

The loss of repetitive discharge in response to a stationary stimulus develops progressively during ALS. In P6–P10 SOD1$^{G93A}$ mice, S-type motoneurons are hyperexcitable (reduced rheobase) whereas F-type motoneurons display a normal excitability (*Leroy et al., 2014*). From P30 to P60, 35% of the motoneurons lose their ability to discharge repetitively and the proportion was even larger at P80 (*Delestrée et al., 2014*) indicating that this loss of function occurs in an increasing number of motoneurons. This loss of ability to discharge repetitively is unrelated to the anesthesia since it has also been found to occur in an in vitro preparation of adult sacrocaudal motoneurons in which no anesthetics were used, and in similar proportion as in lumbar motoneurons (*Delestrée et al., 2014*). In addition, the recruitment current and the F-I gain of the firing motoneurons were unaffected by the mutation both in sacrocaudal motoneurons in vitro (no anesthetics) and in lumbar motoneurons of anaesthetized mice (*Delestrée et al., 2014*). The loss of ability to discharge repetitively, which specifically occurs in the most vulnerable motor units as demonstrated in the present study, is obviously a form of hypoexcitability. A similar time-dependent transition between hyperexcitability to hypoexcitability has been documented in motoneurons derived from ALS patient iPSCs. Four weeks after differentiation, patient-derived motor neurons with the SOD1$^{A4V}$ mutation are hyperexcitable (more action potentials in response to a slow ramp depolarization) (*Wainger et al., 2014*). However, loss of repetitive firing was observed 7–9 weeks after differentiation in patient-derived motor neurons with a SOD1 or a FUS mutation (*Naujock et al., 2016*) or with a C9ORF72 repeat expansion (*Sareen et al., 2014*). Indeed, *Devlin et al. (2015)* showed that TARDBP or C9ORF72 cell lines transitioned over time from hyperexcitable (higher slope of the F-I function at two weeks post-plating) to hypoexcitable (inability to discharge a train starting from 7 to 8 weeks and proportion of hypoexcitable cells increasing later on). The functional impact of the inability to discharge repetitively depends on the pattern of inputs received by the motoneuron in the behaving animal. Large motor units are recruited during brief, powerful movements rather than during postural tasks (*Heckman and Enoka, 2012*). Their preferred input is therefore probably transient rather than stationary. Since we have shown that these motoneurons can still elicit at least one spike during transient inputs, it might be that hypoexcitable motoneurons can still exert a relatively normal function. Together with compensatory mechanisms at the central and peripheral level (*Hadzipasic et al., 2016*; *Quinlan et al., 2017*), this might explain why no obvious motor symptoms are visible before the denervation begins.

## Mechanisms underlying the loss of function

In mouse motoneurons, it has been suggested that loss of repetitive firing could be caused by an insufficient amount of sodium current, and most particularly of its persistent component (*Delestrée et al., 2014*), which is essential for the repetitive firing of spinal motoneurons (*Harvey et al., 2006*; *Kuo et al., 2006*). Recently, *Huh et al. (2017)* successfully performed single-electrode voltage clamp recordings in vivo in pentobarbital-anesthetized adult WT mouse motoneurons. They showed that, although the persistent inward current ('PIC', which includes the persistent sodium current) was generally strong in these motoneurons, the few motoneurons that were unable to fire repetitively were characterized by a large input conductance (i.e. large size, similarly to the present work) and a relatively small PIC. Therefore, these motoneurons had a much smaller PIC relative to their size compared to motoneurons that were able to fire repetitively. This work strongly supports the hypothesis that the inability to sustain repetitive firing is due to a deficit in persistent currents in the motoneurons from ALS mice. In motoneurons derived from ALS patient iPSCs, the

late hypoexcitability has also been associated with an imbalance in the sodium and potassium currents, either a lower amount of sodium current or a larger amount of potassium current or both together (*Sareen et al., 2014*; *Devlin et al., 2015*; *Naujock et al., 2016*). One cause for this imbalance might be a dysregulation of sodium alpha and beta subunits, or voltage-gated potassium channels, which have been demonstrated in iPSC-derived motoneurons from ALS patients (*Sareen et al., 2014*; *Naujock et al., 2016*), suggesting that ALS might involve channelopathy-like mechanisms. Membrane depolarization may also contribute to the ion current imbalance by increasing sodium inactivation and potassium activation. In motoneurons derived from TARDBP or C9ORF72 iPSC lines, the resting membrane potential was found to be more depolarized than in control lines (*Devlin et al., 2015*). The present work shows that the resting membrane potential is, on average, slightly more depolarized in non-repetitively firing cells, in both SOD1$^{G93A}$ and FUS$^{P525L}$ mice, although some of the cells that retain a normal firing were equally depolarized. Our results are somewhat opposed to a recent study in adult slices (*Hadzipasic et al., 2014*), in which the authors identified a population of motoneurons that was assumed to contain a majority of F-type motoneurons, and which displayed a slight hyperpolarization. However, the impact of this hyperpolarization on excitability remained uncertain, as the values of rheobase were not reported. Moreover, the motoneurons that did not fire repetitively in response to a stationary stimulus were excluded from analysis in this study since their classification required repetitive firing. Finally, since this work was conducted in slices, the authors were unable to check the functionality of the neuromuscular junctions of the recorded motoneurons in order to ascertain whether the effects they observed occurred before or after the denervation of the neuromuscular junctions.

## Membrane hypoexcitability versus membrane hyperexcitability?

The membrane depolarization reported in (*Devlin et al., 2015*); and this work) is consistent with the theoretical work by *Le Masson et al. (2014)* suggesting that reduced ATP availability in ALS can result in a chronic membrane depolarization. In their model, *Le Masson et al. (2014)* found that the membrane depolarization was the largest at the axon distal extremity and that it returned closer to normal when going towards the soma. This difference in membrane polarization along the axon offers a potential way to reconcile our work with the threshold tracking technique applied to a peripheral nerve, which often revealed a distal hyperexcitability in ALS patients (*Bostock et al., 1995*). This might occur despite the fact that the cell body of the motoneurons has lost the ability to fire repetitively and can thus be considered hypoexcitable.

## Changes of membrane excitability and degeneration?

*Le Masson et al. (2014)* suggested that decreasing firing rate may prevent an energy crisis due to disruption of ATP production. If degeneration is indeed caused by a disruption in energy homeostasis, loss of repetitive activity could then be a compensatory mechanism to prevent degeneration. This is particularly acute in the largest motoneurons, which require more energy than the smallest ones due to their large membrane surface and larger transmembrane ionic currents (*Attwell and Laughlin, 2001*; *Sengupta et al., 2013*; *Le Masson et al., 2014*). On the other hand, evidence is building to suggest that abnormal intrinsic excitability of motoneurons is by itself harmful in ALS. Retigabine, a specific opener of Kv7 channels—applied at an early stage, when motoneuron derived from ALS patient iPSCs are hyperexcitable—increased their survival (*Wainger et al., 2014*). In contrast, late treatment of hypoexcitable, iPSC-derived motoneurons from ALS patient with a potassium channel antagonist (4-AP) induces a decrease in ER stress and apoptosis markers (*Naujock et al., 2016*). In the same line, in vivo chemogenetic experiments in adult SOD1$^{G93A}$ mice suggested that an early increase of motoneuron—induced by expressing the cation-permeable PSAM (*Magnus et al., 2011*) before the onset of ER stress but after the appearance of misfolded SOD1—decreases misfolded SOD burden, abolishes ER stress and, upon protracted activation, negates the unfolded protein response and preserves muscle innervation (*Saxena et al., 2013*). Conversely, reducing motoneuron excitability has opposite effects (*Saxena et al., 2013*), consistent with a model in which hypoexcitability contributes to motoneuron degeneration in ALS. In two, unrelated mouse models of ALS, we have demonstrated selective loss of function in the motoneurons that preferentially degenerate in the disease. These data support the role of hypoexcitability as a general mechanism contributing to motoneuron degeneration in ALS.

# Materials and methods

## Key resources table

| Reagent type (species) or resource | Designation | Source or reference | Identifiers | Additional information |
|---|---|---|---|---|
| Strain, strain background (Mus musculus B6SJL) | SOD1-G93A | PMID:8209258 | RRID:IMSR_JAX:002726 | |
| Strain, strain background (Mus musculus C57BL/6) | FUS-P525L; FUS-WT | PMID:26842965 | | |
| Gene (Mus musculus) | Chondrolectin | PMID:20437528 | NM_139134.3 | |
| Antibody | anti-digoxigenin alkaline phosphatase-conjugated antibody | PMID:25313866 | RRID:AB_514497 | 1:5000 |
| Antibody | Cy2-conjugated Streptavidin antibody | PMID:25313866 | RRID:AB_2337246 | 7.5 µg/mL |
| Antibody | Neurobiotin | PMID:29256865 | RRID:AB_2336606 | 2% in KCl 3M |
| Antibody | Dextran-TMR | PMID:14566947 | Invitrogen Cat# D1817 | 2% in KCl 3M |
| Antibody | Dextran-FITC | PMID:14566947 | Invitrogen Cat# D1820 | 2% in KCl 3M |
| Antibody | guinea-pig anti-VAChT | PMID:24094105 | RRID:AB_10893979 | 1:500 |
| Antibody | rabbit anti-Phospho-eIF2α | PMID:28131822 | RRID:AB_330951 | 1:50 |
| Antibody | mouse anti-p62 | PMID:28941811 | RRID:AB_945626 | 1:200 |
| Software, algorithm | ImageJ | PMID:22930834 | RRID:SCR_003070 | |
| Software, algorithm | Spike2 | ced.co.uk | RRID:SCR_000903 | v7.16 |

## Mice

All experiments were performed in accordance with European directives (86/609/CEE and 2010–63-UE) and the French legislation. They were approved by Paris Descartes University ethics committee (authorizations CEEA34.MM.064.12 and 01256.02). For the SOD1 experiments, 36 B6SJL-Tg (SOD1*G93A)1Gur/J (SOD1$^{G93A}$; The Jackson Laboratory, line 002726) and 45 of their non-transgenic littermates (WT), aged between 46 and 60 days old (WT: 54 ± 5 days; SOD1$^{G93A}$: 52 ± 5 days) were used. In our colony, SOD1$^{G93A}$ animals have an average lifespan of 127 ± 7 days (N = 18)(end-stage defined as the time when the animals is unable to flip over in less than 30 s when placed on its back). For the FUS experiments, τ$^{ON}$FUS$^{P525L}$ and τ$^{ON}$FUS$^{WT}$ mice were split in two groups: 15 FUS$^{P525L}$ and 10 FUS$^{WT}$ were studied at P30 (mutant: 32 ± 2 days vs. WT: 33 ± 2 days), while 26 FUS$^{P525L}$ and 34 FUS$^{WT}$ were studied at P180 (mutant: 182 ± 3 days vs. WT: 182 ± 3 days). Neither the FUS$^{P525L}$ nor the FUS$^{WT}$ mice showed reduced lifespan (*Sharma et al., 2016*).

## Surgical procedure

The surgical procedure has been described previously (*Manuel and Heckman, 2011*). Briefly, atropine (0.2 mg/kg) mixed in perfusion solution (4% glucose, 1% nAHCO$_3$, 14% Gelatin, pH 7) was given subcutaneously at the onset of the experiment to prevent salivation. After ten minutes, anesthesia was administered intraperitoneally with sodium pentobarbital (70 mg/kg). After confirming absence of noxious reflex, tracheotomy was performed and the mouse was artificially ventilated with 100% oxygen with the SAR-830/AP ventilator (CWE, Ardmore, PA). The end tidal PCO$_2$ was continuously monitored and maintained around 4% (MicroCapstar; CWE). Body temperature was maintained at 37°C using a heating blanket supplemented by an infrared heating lamp and monitored using a rectal temperature probe. The heart rate was monitored (CT-1000; CWE) and maintained between 400 and 500 bpm. A catheter was placed in the jugular vein for supplemental anesthesia (6 mg/kg, every 10–15 min) mixed in perfusion solution. Depth of anesthesia was assessed by a lack of response to noxious stimuli (toe pinch), a stable heart rate, and a stable end-tidal PCO$_2$, showing no signs of resistance against the artificial ventilation. The vertebral columns were immobilized with two pairs of horizontal bars (Cunningham Spinal Adaptor; Stoelting, Dublin, Ireland) applied on the Th12

and L2 vertebral bodies, and the L2–L4 spinal segments were exposed by a laminectomy at the T13–L1 level. A custom made chamber was fitted around the exposed spinal segments and silicon sealant (WPI, Sarasota, FL) was applied to create a recording chamber. This chamber was filled with mineral oil to prevent dehydration of the spinal cord. The nerve innervating either the triceps surae (TS; ankle extensor) or the TA/EDL muscle group (deep peroneal nerve; ankle flexor) was left intact and mounted on a monopolar stimulation electrode. The tendon of the TS or TA/EDL was dissected free and securely attached to an isometric force transducer (BG-100, Kulite). Intracellular recordings were performed using glass microelectrodes (tip diameter 1.0–1.5 μm) filled with KCl 3M (resistance 8–15 MΩ). Intracellular recordings were obtained with an Axoclamp 2B amplifier and Spike2 software (CED, Cambridge, England). All recordings were obtained using discontinuous current clamp (7–9 kHz) and sampled at 20 kHz. At the beginning of the experiment, the initial length of the muscle was set to the length at which the ankle was flexed at 90°. The length was then adjusted in order to record the maximal muscle twitch amplitude. The force signal was amplified (typically 200 or 500x), and filtered at 4 kHz (CyberAmp, Axon Instruments). Surface EMG (i.e. single motor unit compound action potential, MUAP) was recorded at the same time using fine stainless steel wires inserted underneath the muscle fascia (signal amplified 100x, and band-pass filtered at 10 Hz–10 kHz; AM System Model 1700).

## Characterization of the motor unit contractile properties

The twitch properties were recorded in response to single action potentials, and their characteristics measured off-line after averaging at least five sweeps. We measured the twitch amplitude, the contraction time (i.e. the time between the onset of the twitch and the time at which it reached its maximum), the twitch half decay time, and the total duration (i.e. the time between the onset of the twitch and point at which it had declined to 99% of its maximum value). The kinetic parameters of the twitch (contraction time, twitch half decay time and total duration) were strongly correlated. Therefore, we only used the contraction time in our classification procedure.

Fatigue was tested on a long (≥3 min) series of unfused tetanus. The classical fatigue test consists in 13 shocks at 40 Hz, repeated every second (*Burke et al., 1971*). However, when we applied that test to mouse motor units, we observed that, in many motor units (regardless of the genotype), each soma-generated spike was activating fewer and fewer neuromuscular junctions (as indicated by a decrease in the amplitude of the motor unit compound action potential). We therefore decreased the number of stimulation in the train to 8 spikes at 40 Hz, repeated every second (*Figure 1A$_2$–B$_2$–C$_2$*), for at least three minutes. However, even with this reduced number of stimulations per train, we sometimes observed that the MUAPs decreased during the train (*Figure 1A$_2$* filled arrowhead). In many cases, nonetheless, the amplitude of the first MUAP in each train was of constant amplitude, indicating that the same number of neuromuscular junctions was activated for that first stimulation of the train (*Figure 1A$_2$* empty arrowhead). Therefore, we used the amplitude of the first twitch in the train to estimate the fatigability of the muscle fibers. Fatigue index was calculated as the ratio of the amplitude of the first twitch in the train recorded after 3 min of stimulation over the amplitude of the first twitch of the first train. In some cases, even the amplitude of the first MUAP of the train was decreasing over time, and we could not measure of the fatigability of these units. However, we could not identify a particular cause for these failures. We could record units with failing MUAPs and some with stable MUAPs in the same experiment, sometimes one just after the other. We observed these failures in both non-transgenic and transgenic animals, mutant or non-mutant. Besides their inability to sustain muscle fiber activation, these units were indistinguishable from other units and were therefore kept for analysis, despite the fact that their fatigue index could not be measured.

## Measurement of motoneuron excitability

All motoneurons included were identified by antidromic stimulation of their axons and their ability to generate force. All motoneurons retained for analysis had a resting membrane potential more hyperpolarized than −50 mV and an overshooting action potential. Upon penetration, a series of small-amplitude square current pulses were used to plot the I–V relationship. The input conductance was measured from the peak responses (*Manuel et al., 2009*). The discharge properties were investigated using triangular ramps of current (rate 0.1–5 nA/s). F-I curves were obtained by plotting the instantaneous firing frequency versus the intensity of the injected current at the time of the spike.

The voltage threshold for firing was determined on the first spike of a current ramp, as the point at which the first derivative of the voltage reached 10 mV/ms (*Sekerli et al., 2004*). The onset current was determined as the intensity of the current ramp at this time point. The slope of the F-I curve was measured as the slope of a linear regression performed over the 'primary range', when it was present. Motoneurons that responded to the current ramp with a strong discharge were considered to be firing repetitively. Some motoneurons did not discharge at all in response to the current ramp, or only generated 2–3 spikes, despite being able to generate a single overshooting spike when injecting a short square pulse of current. These motoneurons were classified as 'non-repetitively-firing'.

## Chondrolectin expression

In a separate series of experiments, we recorded 19 motoneurons innervating the TS muscle from 19 B6SJL non-transgenic mice (Janvier labs). After characterizing the contractile properties of the motor unit (see above), we injected Neurobiotin (2% in 3M KCl; Vector Laboratories Cat# SP-1120–20, RRID:AB_2336606; iontophoretic injection: 1–5 nA 150 ms pulses at 3.33 Hz for $\geq$5 min) through the microelectrode. Only one motoneuron was labeled per experiment. 45–60 min later, the mouse was perfused intracardially with PBS followed by 4% PFA. After extracting the spinal cord, we performed chondrolectin in situ hybridization as described in *Enjin et al. (2010)*. Chondrolectin probes (Genebank number NM_139134.3) were produced from commercial cDNA (Source BioScience, Nottingham, UK), using T3 RNA polymerase in the presence of digoxigenin-11-UTP (Roche Diagnostics, Basel, Switzerland). Slices were washed with PBT (PBS supplemented with 0.1% Tween-20, Sigma-Aldrich) followed by treatment with 0.5% Triton X-100. Slices were then post-fixed in 4% formaldehyde followed by prehybridization in hybridization buffer (50% formamide, 5 $\times$ saline sodium citrate [SSC], pH 4.5, 1% sodium dodecyl sulphate [SDS], 10 mg/mL tRNA [Life Technologies, Carslbad, CA], 10 mg/mL heparin [Sigma–Aldrich] in PBT). The probe (300 ng/mL) was heat-denatured before starting the overnight hybridization (20–22 hr) at 63°C. Overnight hybridization was followed by sequential washes with wash buffer 1 (50% formamide, 5 $\times$ SSC, pH 4.5% and 1% SDS in PBT) followed by buffer 2 (50% formamide, 2 $\times$ SSC, pH 4.5, and 0.1% Tween-20 in PBT) at 63°C to remove unbound probe. Slices were then washed in 0.1% Tween-20 Tris-buffered saline followed by incubation in 1% blocking reagent (Roche Diagnostics). Then the slices were incubated overnight at 4°C with 1:5000 diluted sheep anti-digoxigenin Fab fragments antibody (Roche Cat# 11093274910, RRID:AB_514497). Hybridized probes were visualized using SIGMAFAST Fast Red TR/Naphthol AS-MX (Sigma–Aldrich). After hybridization, neurobiotin was revealed by washing the slices in PBS followed by PBS-T-G (PBS, 0.25% Triton X-100, 0.25% Gelatin). Slices were then incubated with 7.5 μg/mL Cy2-conjugated Streptavidin antibody (Jackson ImmunoResearch Labs Cat# 016-220-084, RRID: AB_2337246) diluted in PBS-T-G for 2 hr at room temperature. Images were taken on a Zeiss LSM 710 confocal microscope.

## Disease markers immunostaining and image analysis

Electrophysiological recordings were performed in 6 SOD1$^{G93A}$ mice, aged between 46 and 53 days old (50 $\pm$ 2 days; N = 6). 1–2 motoneurons were intracellularly labeled using electrodes filled with KCl 3M + neurobiotin 2% (Vector Laboratories Cat# SP-1120–20, RRID:AB_2336606) and 2% of either Dextran-FITC (Invitrogen Cat# D1820), or Dextran-TMR (Invitrogen, Carlsbad, CA, USA, Cat# D1817). Only one motoneuron was labeled with each color. Soon after the end of the recording, mice were transcardially perfused with 25 mL of ice-cold PBS followed by 2.5 mL/g of 4% PFA in PBS (freshly prepared). At the end of the perfusion-fixation, spinal cords were dissected, post-fixed in 4% PFA in PBS for 18 hr at 4°C and cryoprotected in 30% sucrose in PBS. Spinal cord embedded in OCT were sectioned in cryostat at the thickness of 40 μm. Floating sections were examined in epifluorescence microscopy in order to identify the section(s) containing the labeled motoneuron(s) and to localize the labeled motoneuron(s); identified sections were transferred in a dedicated well and the fluorochrome was photo-bleached by long-duration illumination ($\geq$1 hr, complete bleaching was confirmed by visual examination in epifluorescence microscopy) in order to retain only the neurobiotin labeling in the following multicolor immunostaining procedure. The selected sections were thereafter incubated for 2 hr in blocking buffer (3% BSA +5% Donkey Serum +0.3% Triton-100) at 21°C and then incubated with following primary antibodies: guinea-pig anti-VAChT (1:500; Synaptic Systems Cat# 139 105, RRID:AB_10893979), rabbit anti-Phospho-eIF2$\alpha$ (1:50; Cell Signaling Technology

Cat# 9721, RRID:AB_330951), mouse anti-p62 (1:200; Abcam Cat# ab56416, RRID:AB_945626) and Cy2-conjugated streptavidin (Jackson ImmunoResearch Labs Cat# 016-220-084, RRID:AB_2337246) diluted in blocking buffer, at 4°C for 48 hr under continuous shaking. At the end of the primary antibodies incubation, spinal cord sections were washed in PBS (3 × 45 min) and incubated for 2 hr at 21°C in the mix of appropriate secondary antibodies (Alexa405-conjugated donkey anti-guinea-pig, Alexa568-conjugated donkey anti-rabbit, Alexa633-conjugated donkey anti-mouse, 1:500 diluted in blocking buffer); after an additional round of washing (3 × 45 min in PBS), sections were air-dried and mounted in Fluoroshield mounting medium (Sigma–Aldrich).

Imaging was performed using an LSM710 inverted confocal microscope fitted with a 20 × air objective; settings of the laser power, detector gain and digital gain were adjusted to avoid pixel saturation in the region of interest while obtaining non-zero values for every pixel. Images were acquired at 1024 × 1024 resolution at 8-bit depth (allowing for intensity arbitrary units scale ranging from 0 to 255); optical section thickness was set at 1 µm and confocal stacks of 20–25 optical sections per ROI were acquired. Each fluorescent channel was acquired independently to avoid fluorescence cross-bleeding. Imaging settings were kept constant across multiple samples and imaging sessions.

Image analysis was performed in ImageJ (RRID:SCR_003070; *Schneider et al., 2012*). Image analysis was performed by an operator blind to the electrophysiological features of the imaged motoneuron. Contrast was enhanced for optimal display; the same contrast and brightness values were applied to all images. For the analysis of p-eIF2α fluorescence intensity, 10 optical sections encompassing the target motoneuron (identified by neurobiotin labeling) were collapsed in maximum-intensity projection mode; the contour of each α-motoneuron (identified by the positive cytoplasmic VAChT staining and by the presence of VAChT +C boutons surrounding the cell body) was manually traced using the VAChT channel as reference, and thereafter the mean grey value for the fluorescence intensity in the p-eIF2α channel was computed (in arbitrary units, ranging 0–255); the local background was calculated based on multiple ROI located nearby the motoneuron and subtracted; p-eIF2α staining intensity was computed both for the neurobiotin-labeled motoneuron (NB+) and for nearby NB− motoneurons.

For the quantification of the p62 aggregates burden, 8–10 optical sections spanning the target motoneuron were collapsed in maximum-intensity projection mode. Aggregates were identified as round-shaped inclusions, strongly immunopositive for p62; the quantification of p62 aggregates burden was performed by computing the ratio of the area of p62 aggregates (images were thresholded at the value of 140 arbitrary units [within the 8-bits range 0–255] and only the area brighter than the threshold was measured) over the total surface of the section of the cell (measured by manually tracing the contour of the cell in the VAChT channel). The value of the p62 burden was expressed as percentage of the cell area occupied by aggregates.

## Statistical analysis

All data were analyzed in Spike2 (CED, RRID:SCR_000903) and IPython (RRID:SCR_001658; *Perez and Granger, 2007*). Statistical tests were conducted using the libraries SciPy v.0.19.1 (RRID: SCR_008058; *Jones et al., 2001*) and statsmodel v.0.8.0 (SCR_016074; *Seabold and Perktold, 2010*). In all cases, results were considered statistically significant if $p < 0.05$. Data are reported as mean ±SD, along with sample size (number of motoneurons, unless stated otherwise) in every case. No test was performed to detect outliers, and no data points were excluded from analysis. Comparisons between two independent samples were conducted using a two-tailed Mann-Whitney U-test. Contingency tables were analyzed using either the Fisher exact test (for 2 × 2 tables) or a $\chi^2$ test (for 3 × 2 tables). Cases where the effect of two independent variables was considered were analyzed using two-way ANOVA. Normality of the residuals was tested using the Shapiro-Wilk test, and homoscedasticity of the datasets was tested using Levene's test. The twitch amplitude was the only parameter that had to be log-transformed to satisfy the assumptions of ANOVA. None of the tests reported here yielded a significant interaction term and therefore only the main effects are reported. Linear regression slopes were compared using ANCOVA.

## Acknowledgements

This work has benefited from the support and expertise of the microscopy platform and the animal facility of the Saints-Pères faculty of biomedical sciences at Paris Descartes University. We would like to thank Dr. Boris Lamotte d'Incamps for careful reading of the manuscript, and Martyn Goulding and his lab for precious help with in situ hybridization. This work was financed by NIH-NINDS R01NS077863, TARGET-ALS, the Association Française contre les Myopathies (AFM) project 'HYPERTOXIC', and Thierry Latran Foundation. FR was supported by the Synapsis Foundation and by the Baustein program of Ulm University Medical Faculty.

## Additional information

### Funding

| Funder | Grant reference number | Author |
| --- | --- | --- |
| Consejo Nacional de Ciencia y Tecnología | CONACYT | María de Lourdes Martínez-Silva |
| Target ALS | | Aarti Sharma<br>Neil A Shneider<br>Daniel Zytnicki<br>Marin Manuel |
| National Institute of Neurological Disorders and Stroke | R01NS077863 | CJ Heckman<br>Marin Manuel |
| Stiftung Synapsis - Alzheimer Forschung Schweiz AFS | | Francesco Roselli |
| Ulm University | Baustein Program | Francesco Roselli |
| AFM-Téléthon | HYPERTOXIC | Daniel Zytnicki<br>Marin Manuel |
| Fondation Thierry Latran | OHEX | Marin Manuel<br>Daniel Zytnicki |

The funders had no role in study design, data collection and interpretation, or the decision to submit the work for publication.

### Author contributions

María de Lourdes Martínez-Silva, Formal analysis, Investigation, Visualization, Writing—original draft, Writing—review and editing; Rebecca D Imhoff-Manuel, Data curation, Formal analysis, Investigation, Visualization; Aarti Sharma, Neil A Shneider, Data curation, Formal analysis, Writing—review and editing; CJ Heckman, Conceptualization, Supervision, Funding acquisition, Writing—review and editing; Francesco Roselli, Data curation, Formal analysis, Investigation, Methodology, Writing—review and editing; Daniel Zytnicki, Supervision, Funding acquisition, Writing—original draft, Project administration, Writing—review and editing; Marin Manuel, Conceptualization, Data curation, Software, Formal analysis, Supervision, Funding acquisition, Investigation, Visualization, Methodology, Writing—original draft, Project administration, Writing—review and editing

### Author ORCIDs

Neil A Shneider http://orcid.org/0000-0002-3223-7366
Daniel Zytnicki http://orcid.org/0000-0002-0431-9604
Marin Manuel http://orcid.org/0000-0002-5344-3572

### Ethics

Animal experimentation: All experiments were performed in accordance with European directives (86/609/CEE and 2010-63-UE) and the French legislation. They were approved by Paris Descartes University ethics committee (authorizations CEEA34.MM.064.12 and 01256.02). All surgery was performed under sodium pentobarbital anesthesia, and every effort was made to minimize suffering.

Decision letter and Author response
Decision letter https://doi.org/10.7554/eLife.30955.018
Author response https://doi.org/10.7554/eLife.30955.019

## Additional files

### Supplementary files

• Transparent reporting form
DOI: https://doi.org/10.7554/eLife.30955.016

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
