## [Decision Letter]

Thank you for submitting your article "Hypoexcitability precedes denervation in the large fast-contracting motor units in two unrelated mouse models of ALS" for consideration by *eLife*. Your article has been reviewed by three peer reviewers, and the evaluation has been overseen by a Reviewing Editor and David Van Essen as the Senior Editor. The following individuals involved in review of your submission have agreed to reveal their identity: Brian Wainger (Reviewer #2); Bruce P Bean (Reviewer #3).

The reviewers have discussed the reviews with one another and the Reviewing Editor has drafted this decision to help you prepare a revised submission.

Summary:

This manuscript presents new and potentially important data challenging an observation that has become completely accepted by most investigators: that motor neuron hyperexcitability underlies and precedes motor neuron death in diseases such as ALS. In fact, many in the neuroscience/neurology community believe that hyperexcitability may be "warning sign" associated with a variety of neurodegenerative conditions. All of the reviewers think this paper is significant and should be published in *eLife* after some additional experiments are carried out or after additional information is provided. We have summarized below what we regard as the key points from each of the reviewers. As you will see, most of the reviewers asked for additional information concerning properties of the different types of motor neurons. Reviewer 2 asks for a separate, morphological, confirmation to confirm the state of degeneration of the susceptible motor neurons.

Essential revisions:

*Reviewer 1:*

1) More mechanistic information especially a more detailed characterization of the electrophysiological properties of the different populations of motor neurons should be provided.

2) More recordings from the S population of motor neurons must be included.

Reviewer 2:

1) The authors should present more information on the coupling between hypo/hyper excitability and motor neuron degeneration using additional methods such as immunocytochemistry to ask whether motor neurons have undergone functional decline even though they can still elicit muscle contraction.

2) More data on motor neuron membrane potential should be included.

3) The authors should carry out additional characterization of FF motor neurons in the SOD mice.

*Reviewer 3:*

1) The authors should discuss the possible effects of anesthesia on motor neuron properties.

2) More information on the age of the SOD mice used in this study (as well as their average lifetimes in the authors' lab) should be given.

---

## [Author Response]

Summary:This manuscript presents new and potentially important data challenging an observation that has become completely accepted by most investigators: that motor neuron hyperexcitability underlies and precedes motor neuron death in diseases such as ALS. In fact, many in the neuroscience/neurology community believe that hyperexcitability may be "warning sign" associated with a variety of neurodegenerative conditions. All of the reviewers think this paper is significant and should be published in eLife after some additional experiments are carried out or after additional information is provided. We have summarized below what we regard as the key points from each of the reviewers. As you will see, most of the reviewers asked for additional information concerning properties of the different types of motor neurons. Reviewer 2 asks for a separate, morphological, confirmation to confirm the state of degeneration of the susceptible motor neurons.

We have carried out many additional experiments in order to revise the paper according to the reviewer’s recommendations. We have investigated supplemental motor units in order to increase the number of Slow-type motoneurons from the ankle extensor motor pool. It should be noted that these motoneurons are scarce among the total population of motoneurons and also the smallest in size. They are therefore the most difficult to record in vivo. Nevertheless, we investigated 34 additional motor units from SOD1^G93A^ mice in order to get 5 extra S-type motoneurons (the others supplemental motoneurons were 21 FR motoneurons and 8 FF motoneurons). In addition, we provided a more detailed characterization of the electrophysiological properties of the different types of motoneurons. We also carried out experiments in which we intracellularly labelled with neurobiotin eight motoneurons that have been electrophysiologically investigated. The expression of two disease markers (p-eIf2a and p62 aggregates) were subsequently studied in these motoneurons using immunocytochemistry methods. We have added a full section in the Results part (“The loss of electrophysiological function represents a more advanced stage of disease progression”) to describe these new data.

The manuscript has been carefully revised in order to include all the additional data, information and revisions as recommended by the reviewers (see below for a point by point rebuttal).

Finally, please note that we received significant help from three colleagues (Francesco Roselli, Aarti Sharma and Neil Shneider) in the revision process of the manuscript, including the new experiments that we performed. Their contributions justify their inclusion in the author list (the change of authorship has been acknowledged by all the contributing authors).

Essential revisions:Reviewer 1:1) More mechanistic information especially a more detailed characterization of the electrophysiological properties of the different populations of motor neurons should be provided.

We now provide a more detailed characterization of the electrophysiological properties of different populations of motoneurons, in both SOD1 and FUS mouse models. These data are provided in Tables 2 and 4. We now mention these data and refer to the tables in the Results section wherever appropriate.

We’ve expanded the section “Mechanisms underlying the loss of function” of the Discussion to better highlight a recent work that provides a plausible mechanistic explanation for the loss of repetitive firing.

2) More recordings from the S population of motor neurons must be included.

We’ve performed experiments on 11 additional SOD1 mice. Given the small proportion of S-type units in the TS muscle (Bloemberg and Quadrilatero, 2012), we had to record 34 additional units in order to record 5 extra S motoneurons. The other motoneuron include 15 classified as small FR, 6 as large FR and 8 FF. Overall, these new data bring the total SOD1 population to 15 FF (16%), 65 FR (small and large together, 71%) and 12 S (13%) motor units. These proportions are in line with the expected proportions of motoneurons in Gastrocnemius+Soleus motor pool (Bloemberg and Quadrilatero, 2012). All statistics and figures have been updated to reflect this new sample size.

These new data reinforce our conclusion that the loss of repetitive firing is specific of FF and large FR motor units.

Reviewer 2:1) The authors should present more information on the coupling between hypo/hyper excitability and motor neuron degeneration using additional methods such as immunocytochemistry to ask whether motor neurons have undergone functional decline even though they can still elicit muscle contraction.

This is a very interesting question, one that would probably deserve a paper on its own. Nonetheless, we’re providing new data that, we think, help shed some light on this point. Doing so was very challenging. We performed 16 additional experiments. We’ve developed new protocols for labeling several motoneurons by experiments, without sacrificing precious spectral space for subsequent multicolor immunolabelling: we’re co-injecting a fluorescent dye together with Neurobiotin. The fluorescent dye allows us to identify the location of the labeled motoneuron in a large number of spinal cord sections, then the dye is photobleached and the Neurobiotin from the – possibly several – labeled motoneurons is reveled with a single fluorescent dye (Cy2). Despite this more efficient approach, we had to use part of the spectrum for VAChT immuno to measure disease markers in the remaining population of motoneurons, leaving only 2 channels open to test at most 2 disease markers.

Despite all the points at which this whole process could fail, we managed to study eight motoneurons, in which we could record both the electrical activity and the presence or absence of disease markers by immunohistochemistry, from six SOD1-G93A mice. The expression of two disease markers (p-eIf2a and P62 aggregates) were subsequently studied in these motoneurons using immunocytochemistry methods. We have added a full section in the Results part (“The loss of electrophysiological function most likely represents a more advanced stage of disease progression”) to describe these new data.

2) More data on motor neuron membrane potential should be included.

We have now added data from 34 additional motor units (see above). We now provide more information on the resting membrane potential of the motoneurons in each physiological type, as well as separated in large and small units. These data are provided in Tables 2 and 4.

3) The authors should carry out additional characterization of FF motor neurons in the SOD mice.

Among the newly recorded motoneurons, we have recorded 8 additional FF motoneurons. We provided a more thorough characterization of the electrical and mechanical properties of the different types of motoneurons (including FFs) in Tables 1–4.

Reviewer 3:1) The authors should discuss the possible effects of anesthesia on motor neuron properties.

This point is now addressed in the Discussion part (“Loss of function”).

2) More information on the age of the SOD mice used in this study (as well as their average lifetimes in the authors' lab) should be given.

The age of the mice was buried in the Materials and methods. We are now clearly stating the age of the mice used at the beginning of the Results section. The average lifespan for the SOD1-G93A mice is now stated in the Materials and methods section.